# ATTENTION IS ALL YOU NEED FOR MIXTURE-OF-DEPTHS ROUTING

## ABSTRACT

Advancements in deep learning are driven by training models with increasingly larger numbers of parameters, which in turn heightens the computational demands. To address this issue, Mixture-of-Depths (MoD) models have been proposed to dynamically assign computations only to the most relevant parts of the inputs, thereby enabling the deployment of large-parameter models with high efficiency during inference and training. These MoD models utilize a routing mechanism to determine which tokens should be processed by a layer, or skipped. However, conventional MoD models employ additional network layers specifically for the routing which are difficult to train, and add complexity and deployment overhead to the model. In this paper, we introduce a novel attention-based routing mechanism *A-MoD* that leverages the existing attention map of the preceding layer for routing decisions within the current layer. Compared to standard routing, *A-MoD* allows for more efficient training as it introduces no additional trainable parameters and can be easily adapted from pre-trained transformer models. Furthermore, it can increase the performance of the MoD model. For instance, we observe up to $2\%$ higher accuracy on ImageNet compared to standard routing and isoFLOP ViT baselines. Furthermore, *A-MoD* improves the MoD training convergence, leading to up to $2\times$ faster transfer learning.

## 1 INTRODUCTION

Increasing the model size has enabled transformer-based deep learning models to achieve state-of-the-art performance across various domains, including computer vision (Dosovitskiy et al., 2021) and natural language processing (Hoffmann et al., 2022; Kaplan et al., 2020) – even unlocking emergent capabilities (Wei et al., 2022). However, the computational costs of these large models present significant challenges (Thompson et al., 2020). Therefore, reaching a Pareto-optimal model to maximize both efficiency and performance is crucial.

Jacobs et al. (1991) originally introduced conditional computation via mixture of experts, laying the foundations to increase model sizes while maintaining FLOPs, by dynamically activating only a subset of the model parameters, termed experts, conditioned on the input. This principle allowed scaling towards outrageously large networks (Shazeer et al., 2016) and is leveraged at the forefront of current Large Language Models (LLMs) (Jiang et al., 2024).

Compared to standard deep learning models Dosovitskiy et al. (2021); Wang et al. (2024); He et al. (2016), dynamic models have received less research attention and are often not yet competitive on the Pareto front of performance and runtime on standard GPU architectures. Here, we focus on further advancing the field of dynamic compute.

Recently, Raposo et al. (2024) introduced Mixture-of-Depths (MoD) as a variant of mixture of experts. In MoD models, the computational costs are dynamically reduced by processing only a subset of tokens in a layer while the remaining tokens skip the layer (see Fig. 2a). Compared to baselines with equivalent FLOPs, MoDs can perform favorably on language tasks. A crucial component of MoD is its router, which receives tokens as inputs and, given a user-defined capacity, determines which tokens should enter or skip a layer. The router usually consists of a linear layer that is jointly trained along with the model (Fig. 2b).

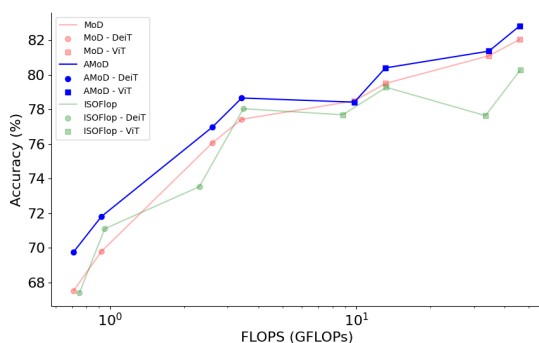

Figure 1: Accuracy vs FLOPs Pareto-curve for *A-MoD* in comparison with MoD and ISOFlop models on ImageNet-1k.

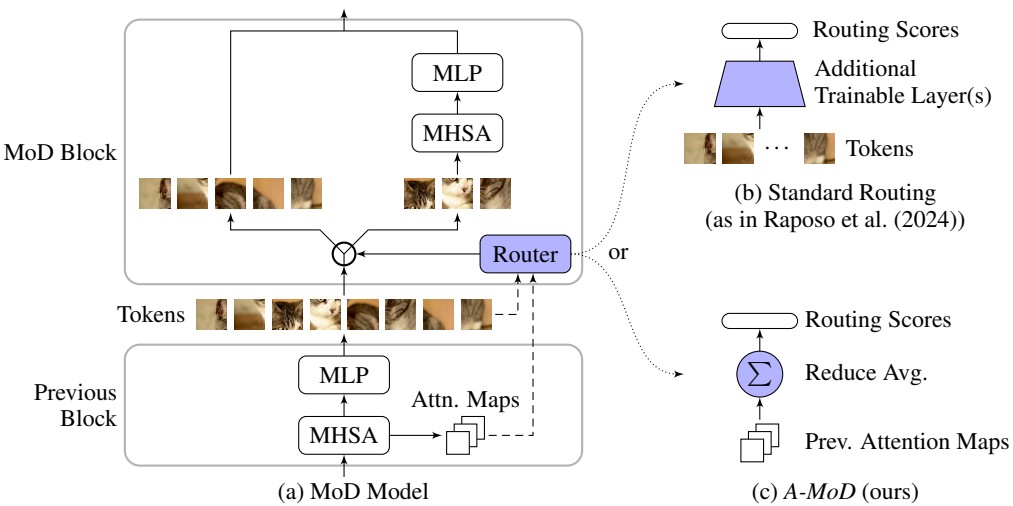

Figure 2: MoD model (a) with standard routing (b) vs. our *A-MoD* attention routing (c).

The routing mechanisms heavily influence the model performance for several reasons. First, routing introduces noise into the training process, as the routing is a discrete decision and is often performed at multiple layers and per token. Second, routers depend on additional layers, and hence, need to be trained from scratch when adapting a vanilla pretrained model to an MoD model. Lastly, the router adds a small computational overhead to the sparsified model.

Hence, in this paper, we ask and address the question: *Can we improve the routing mechanism in MoD models based on information that is already available within the model, instead of using additional trainable parameters within the router?* We find the answer to our question in the attention mechanism of commonly used transformer architectures (Vaswani et al., 2017; Dosovitskiy et al., 2021).

We assume that the attention maps can be used to estimate the importance of a token, by averaging its interaction with other tokens. Based on that, we propose to aggregate the information in the attention maps and use it as an importance measure for token routing in MoD. We call our method attention routing for MoD: *A-MoD* (Fig. 2c). We find that *A-MoD* can outperform standard routing in MoD networks across a range of model sizes and tasks consistently (as shown in Fig. 2). Not only is our *A-MoD* parameter-free, but it can also be applied to adapt off-the-shelf pretrained transformer models to MoDs with almost no additional training. We further validate our method empirically and show that routing scores computed by *A-MoD* are better correlated with token importance estimates compared to routing scores from standard routers.

This paper presents a significant advancement in the application of MoD to the visual domain. Our primary contributions are:

- We find that MoD is not only viable but also advantageous for visual tasks, providing empirical evidence that it can outperform traditional models in terms of both FLOPs and performance.

- We introduce *A-MoD*, a parameter-free routing method for MoDs based on the attention maps to compute token importance.

- We demonstrate that *A-MoD* outperforms a standard router across a range of datasets for MoDs on finetuning and transfer learning. In the case of transfer learning, *A-MoD* exhibits faster convergence of MoD models transferred from a dense pretrained model.

- Compared to standard MoD, *A-MoD* consistently selects important tokens, and routing decisions correlate with leave-one-out token importance that is estimated by removing tokens.

## 2 RELATED WORK

**Attention Maps**    The attention mechanism (Bahdanau, 2014) enables models to learn long-range dependencies within sequences with a constant number of operations. Transformers (Vaswani et al., 2017) leverage the attention mechanism in the language domain and have become a de facto standard model. Dosovitskiy et al. (2021) further adapted transformers to the vision domain by treating image patches as tokens, introducing the vision transformer (ViT). For images, attention maps have been shown to focus on key areas such as objects in the image (Carion et al., 2020a; Jetley et al., 2018). In this paper, we utilize this property for effective routing of tokens within neural networks. Furthermore, we use the Data-efficient image Transformers DeiT-T and DeiT-S (Touvron et al., 2021) instead of vanilla ViT-T and ViT-S models, as small ViTs do not generalize well when trained on smaller datasets (Dosovitskiy et al., 2021).

**Mixture of Experts and Mixture-of-Depths**    Since their introduction over three decades ago (Jacobs et al., 1991; Jordan & Jacobs, 1993), Mixture of Experts (MoE) have been applied to various model types. Shazeer et al. (2016) introduced MoEs to scale transformer architectures (Ludziejewski et al., 2024). Subsequently, MoEs have achieved extensive empirical success across vision and language tasks (Puigcerver et al., 2024; Jain et al., 2024; Fedus et al., 2022a; Riquelme et al., 2021). One of the main challenges when training MoE networks is training instability (Zoph et al., 2022; Fedus et al., 2022b). Raposo et al. (2024) recently introduced the Mixture-of-Depths (MoD) architecture, where each transformer block processes only a subset of tokens, achieving a favorable compute-performance trade-off compared to large transformer models. Liang et al. (2022) also induce sparsity by fusing tokens together without entirely skipping tokens. In their current form, both MoEs and MoDs use dedicated routing networks that decide which components of the overall network process which tokens. The difference between MoEs and MoDs is that an MoE model comprises several distinct experts that independently process the tokens. In contrast, an MoD model only chooses between two experts per layer one of which is the layer itself and the other an identity function, see Fig. 2.

**Routing Methods**    Routing mechanisms are required for most conditional computation blocks (Cai et al., 2024). In MoE models for transformers, the purpose of the router is to match tokens to experts such that performance is maximized. In case of models with a single expert such as Switch Transformers (Fedus et al., 2022b) or MoD, routers decide whether a token will benefit from processing by the expert or will be skipped. Various methods (Liu et al., 2024) have been proposed such as learned routers (Shazeer et al., 2016) with token choice or expert choice routing (Zhou et al., 2022), solving a linear program to match tokens to experts (Lewis et al., 2021), hashing inputs to match experts (Roller et al., 2021) or using reinforcement learning to make routing decisions (Clark et al., 2022; Bengio et al., 2015; 2013). Explicitly learning the routers is the current state-of-the-art that outperforms other methods in most cases (Dikkala et al., 2023). However, this approach mainly proves effective with a larger number of routing parameters and is prone to training instabilities (Ramachandran & Le, 2019). Thus, training routers that consistently lead to strong performance remains an open problem.

Our work focuses on improving the MoD architecture. We propose a novel routing mechanism, based on attention maps, thereby eliminating the need for a standard router. The tokens are routed in a parameter-free manner without any extra computational overhead.

## 3 METHOD

In this section, we explain the Mixture-of-Depths (MoD) architecture and introduce our attention-based MoD routing algorithm, *A-MoD*, that can be employed to improve its routing.

### 3.1 MIXTURE-OF-DEPTHS

Our work focuses on Vision Transformers (Dosovitskiy et al., 2021; Touvron et al., 2021). Here, given an input in terms of tokens $\mathbf{X}$, the output predictions are calculated by a model $f(\mathbf{X}; \Theta)$ consisting of $L$ Transformer blocks parameterized by a set of learnable weights $\Theta$. Each transformer block includes a Multi-Head Self-Attention (MHSA) with $H$ heads, followed by a two-layers fully-connected network with GeLU activations (MLP).

In MoD, Raposo et al. (2024) introduce a variation of transformer-based architectures with the assumption that individual tokens require varying amounts of compute within a model. In particular, MoD layers only process a subset of selected important tokens, while the remaining tokens skip the layer. Empirically, this procedure can improve the performance over a vanilla ViT with a comparable compute budget.

Whether or not tokens skip a layer is determined by token importance scores estimated by a routing algorithm. Conventionally, standard routing computes these importance scores with additional layers (see Section 3.2). In contrast, our *A-MoD* computes the scores directly from the attention maps of previous layers without the need of additional parameters (see Section 3.3).

### 3.2 STANDARD ROUTING

Considering a single MoD layer, the standard approach to compute the importance scores of input tokens requires an additional router network, as shown in Figure 2b. Typically, a router is a linear layer that projects a token vector to a scalar representing its importance score (as introduced by Raposo et al. (2024)). Formally, we consider the $l$-th transformer layer $f_l(\mathbf{X}^{l-1}; \theta_l)$ parameterized by a set of parameters $\theta_l$ with an input $\mathbf{X}^{l-1} = \left[ \boldsymbol{x}_1^{l-1}; \boldsymbol{x}_2^{l-1}; \ldots; \boldsymbol{x}_N^{l-1} \right] \in \mathbb{R}^{N \times d}$ representing a token sequence of length $N$. Now, we can estimate token importance scores as:

$$\boldsymbol{r}_i = (\mathbf{X}^{l-1} \mathbf{W}_r^l)_i, \tag{1}$$

where $\mathbf{W}_r^l \in \mathbb{R}^{d \times 1}$ is the parameter of the additional linear routing network. These tokens will be skipped or processed based on their scores as per the equation below:

$$\boldsymbol{x}_i^l = \begin{cases} r_i f_l \left( \mathbf{X}^{l-1} \right)_i + \boldsymbol{x}_i^{l-1} & \text{if} \quad \boldsymbol{r}_i \geq P_\beta(\mathbf{R}^l) \\ \boldsymbol{x}_i^{l-1} & \text{else} \end{cases} \tag{2}$$

Here, $P_\beta(\mathbf{R}^l)$ denotes the $\beta$-th percentile of all token importance scores $\mathbf{R}^l$. $\beta$ can be defined in terms of the capacity $C$ as $\beta := 1 - \frac{C}{N}$, where $C \in (0,1)$ is the capacity for the MoD layer. To learn the token importance scores during backpropagation, the output of the transformer layer is multiplied by the importance scores $r_i$, such that it can receive a non-zero gradient.

### 3.3 ATTENTION ROUTING

In contrast to standard routing, we propose *A-MoD*, a method to compute routing scores based on attention without additional trainable parameters. *A-MoD* leverages the attention map of the previous layer to determine the routing scores for the current MoD layer, as shown in Figure 2c. The attention map $\mathbf{A}_h^{l-1} \in \mathbb{R}^{N \times N}$ of the $h$-th head from the previous layer can be computed as follows Vaswani

et al. (2017):

$$A_h^{l-1} = \text{softmax}\left(\frac{(Q_h^{l-1})(K_h^{l-1})^T}{\sqrt{d}}\right), \tag{3}$$

where $Q_h^{l-1} \in \mathbb{R}^{N \times d}$ and $K_h^{l-1} \in \mathbb{R}^{N \times d}$ are query and key matrices computed from the previous layer respectively, and $d$ is the embedding dimension of query and key.

Following Equation 3, each element $\boldsymbol{a}_{h,ji}^{l-1}$ of $A_h^{l-1}$ indicates how much information from the $i$-th token is considered when computing the $j$-th output. Aggregating $\boldsymbol{a}_{h,ji}^{l-1}$ across all rows yields a measure for the relevance of the $i$-th token with respect to all other tokens. Therefore, in *A-MoD*, we propose to compute a token importance score by averaging the corresponding attention values across all rows and attention heads as:

$$\boldsymbol{r}_i = \frac{1}{HN} \sum_{h=1}^{H} \sum_{j=1}^{N} \boldsymbol{a}_{h,ji}^{l-1}. \tag{4}$$

Based on the score computation above, the output from the $l$-th layer can then be calculated as:

$$\boldsymbol{x}_i^l = \begin{cases} f_l\left(\mathbf{X}^{l-1}\right)_i + \boldsymbol{x}_i^{l-1} & \text{if} \;\; \boldsymbol{r}_i \geq P_\beta(\mathbf{R}^l) \\ \boldsymbol{x}_i^{l-1} & \text{else} \end{cases} \tag{5}$$

We note that, for *A-MoD*, we do not multiply the token scores $\boldsymbol{r}_i$ by the output, as the attention maps are already learnable in the previous layer. This preserves the original token output, promoting faster training when adapting from a vanilla pretrained checkpoint. We also tried a variation with multiplying $\boldsymbol{r}_i$, but this did not lead to performance improvements and, therefore, was removed in the favor of simplicity. For standard routing in Equation 2, this multiplication term is required to properly calculate the gradient of the router parameters. In contrast, *A-MoD* removes the parameters of the router and thereby enables easier post-hoc adaptation of MoDs and eliminates training instabilities of routing scores.

## 4 EXPERIMENTS

### 4.1 TRAINING SETUP AND OVERVIEW

In our experiments, we systematically evaluate *A-MoD* and empirically demonstrate its benefits over standard routing for MoDs. We perform evaluations across a range of model architectures and multiple image classification tasks. In each experiment, we train a MoD, adapted from a vanilla pretrained transformer model. We conduct experiments on both finetuning the adapted MoD model on the same dataset used for training the vanilla pretrained transformer model and transfer learning on different datasets.

**Training setup** We evaluate *A-MoD* across four vision transformer architectures of varying sizes: DeiT-Tiny, DeiT-Small (Touvron et al., 2021), ViT-Base and ViT-Large (Dosovitskiy et al., 2021). Each MoD architecture is adapted from a vanilla pretrained checkpoint on ImageNet-1k (Russakovsky et al., 2015). Starting from this checkpoint, we train the MoD models with $50\%$ and $12.5\%$ capacity as described in Eq. (5), i.e., $50\%$ and $12.5\%$ tokens are processed in each MoD layer, respectively. Following Raposo et al. (2024), we alternate between MoD layers and dense layers in our MoD architecture i.e. every second layer is an MoD layer. We also analyze the effect of placing MoD layers only in the later layers of the model in Section 4.5.

**Finetuning** We finetune the MoD models on ImageNet-1k. For each case, we compare our *A-MoD* to standard routing. We also compare both MoD variants to an isoFLOP vanilla vision transformer. This isoFLOP model is obtained by appropriately reducing the number of layers of the original model to match the number of FLOPs of its MoD counterpart. Only reducing the layers still allows the isoFLOP model to benefit from the weights of the pretrained checkpoint. Each model is trained with the AdamW optimizer (Loshchilov & Hutter, 2017) for 100 epochs using a batch size of 128 and a learning rate of $1e-5$ with a linear warmup followed by cosine annealing. We identify this learning rate schedule after performing a sweep as shown in Figs. 12(a), 12(b) and 13 in the Appendix.

Table 1: *A-MoD* **mostly outperforms MoD with standard routing and the isoFLOP baseline on ImageNet**, both for $50\%$ and $12.5\%$ capacity.

| Model | Configuration | C= 12.5% | | C= 50% | |
|---|---|---|---|---|---|
| | | FLOPs (G) | Accuracy (%) | FLOPs (G) | Accuracy (%) |
| DeiT-Tiny | isoFLOP | 0.75 | 67.4 | 0.95 | 71.1 |
| | MoD | 0.71 | 67.52 | 0.92 | 69.78 |
| | *A-MoD* | 0.71 | **69.76** | 0.92 | **71.8** |
| DeiT-Small | isoFLOP | 2.3 | 73.53 | 3.47 | 78.04 |
| | MoD | 2.6 | 76.07 | 3.42 | 77.43 |
| | *A-MoD* | 2.6 | **76.98** | 3.42 | **78.66** |
| ViT-Base | isoFLOP | 8.8 | 77.69 | 13.21 | 79.28 |
| | MoD | 9.8 | **78.49** | 13.1 | 79.5 |
| | *A-MoD* | 9.8 | 78.42 | 13.1 | **80.4** |
| ViT-Large | isoFLOP | 33.4 | 77.64 | 46.24 | 80.28 |
| | MoD | 34.5 | 81.1 | 45.92 | 82.04 |
| | *A-MoD* | 34.5 | **81.37** | 45.92 | **82.82** |

**Transfer learning**  To further investigate the benefits of *A-MoD*, we perform transfer learning for image classification on the smaller Stanford Cars (Krause et al., 2013), Oxford Pets (Parkhi et al., 2012) and Flowers102 (Nilsback & Zisserman, 2008) datasets. Here, each model is trained with SGD for 200 epochs, a batch size of 64 and learning rate 0.01 with cosine annealing.

**Token importance**  Finally, we conduct a comparison between the routing scores computed by standard routing and *A-MoD*, with a reference score that measures the importance of each token. This analysis enables us to further distinguish the benefits of *A-MoD*. We use a leave-one-out method (Hastie et al., 2009) to estimate the token importance. In particular, we measure the *change in the loss of the model if a certain token is removed at an MoD layer*. This allows us to assign a reference importance score to each token in each MoD layer, for every input image. We then correlate this with the our routing weights for each MoD layer and token, both, for our attention-based routing and standard routing. Overall, not only does *A-MoD* choose visually relevant tokens, but the routing scores also correlate strongly with the leave-one-out token importance.

### 4.2  *A-MoD* IMPROVES PERFORMANCE FOR FINETUNING

For finetining, we train each MoD model on ImageNet. Across all our considered vision transformer models (ranging from 5M to 300M parameters), *A-MoD* mostly outperforms standard routing. Results for MoDs with $50\%$ and $12.5\%$ capacity are presented in Table 1. Through the training curves presented In Fig. 3 for $50\%$ capacity and Fig. 9 for $12.5\%$ capacity in the Appendix we highlight that *A-MoD* converges faster.

For the DeiT-Tiny model with $50\%$ capacity (see Fig. 3(a)), *A-MoD* outperforms MoD by more than $2\%$ and by $1\%$ on the other larger models. Similarly, for $12.5\%$ capacity, *A-MoD* outperforms standard routing on both DeiT-Tiny and Small and is on par for the larger variants. While *A-MoD* is marginally worse for the ViT-Base model for $12.5\%$ capacity, it requires fewer epochs to converge as shown in the convergence plots in Fig. 9(c) (in the Appendix) and already achieves this peak at the 20-th epoch. Overall, Table 1 along with the training curves in Fig. 3 confirm that *A-MoD* can outperform MoDs with standard routing as well as isoFLOP baselines. Specifically, *A-MoD* has larger performance improvements for the smaller DeiT-Tiny and DeiT-Small and enables faster convergence across all models.

**Adapting from pretrained checkpoints**  As described in Eq. (5), *A-MoD* can compute routing scores solely based on the attention maps and it does not multiply the output of each MoD block with the routing score, thus mostly conserving the token output. Both properties allow *A-MoD* finetuned from a pretrained checkpoint with attention routing to converge with minimal training. Fig. 3 illustrates that *A-MoD* enables much faster convergence, greatly reducing the required training

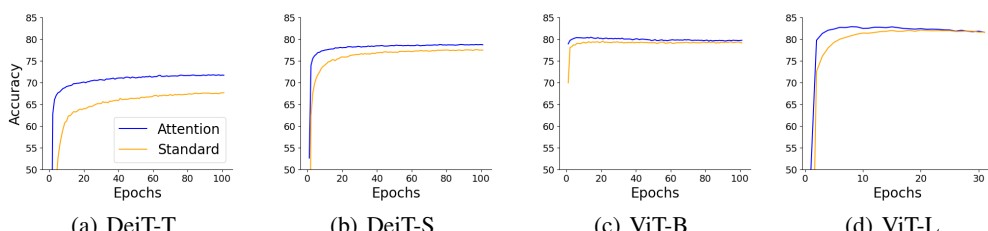

(a) DeiT-T      (b) DeiT-S      (c) ViT-B      (d) ViT-L

Figure 3: **A-MoD achieves better performance and faster convergence on ImageNet-1k.** Fine-tuning with *A-MoD*: Results comparing *A-MoD* with standard routing and isoFLOP baselines with 50% capacity on ImageNet.

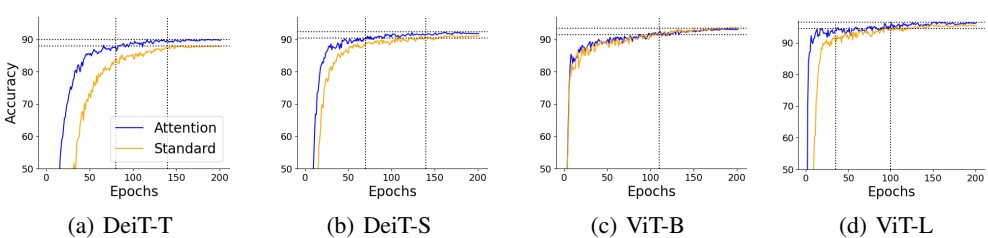

(a) DeiT-T      (b) DeiT-S      (c) ViT-B      (d) ViT-L

Figure 4: **A-MoD converges faster across different datasets** Transfer learning with *A-MoD*: *A-MoD* with 50% capacity MoD trained on the Flower102 dataset. Dotted lines denote the epochs needed to reach within 2% of peak accuracy.

time compared to standard routing. In some cases, *A-MoD* can achieve reasonable accuracy without any training. This is exemplified in Fig. 3(c), where *A-MoD* achieves 78% accuracy without any training. All accuracies of MoDs adapted from a pretrained checkpoint without training are reported in Table 3 in the Appendix and highlight that *A-MoD* always starts from a higher accuracy than standard routing. This is possible as the model estimates the least important tokens using the already learned attention maps, such that final accuracy is minimally affected as further substantiated in Section 4.4. In contrast, standard routing multiplies layer outputs by the routing scores and needs to learn routing from scratch, as it is based on additional layers. These factors result in slower convergence.

**Multiplying routing scores to output in *A-MoD*** In Fig. 11 in the Appendix, we compare *A-MoD* with a modification that multiplies the output of the MoD block with the attention routing score to verify if *A-MoD* benefits from an additional learned gradient like standard routing, i.e., using Eq. (2) instead of Eq. (5). However, multiplying the routing scores to the output for *A-MoD*, instead, worsens accuracy of the adapted MoD model without any training and slightly slows down convergence.

**Learning rate stability analysis** To investigate the stability of our training with respect to the learning rate for *A-MoD* and MoD, we perform a sweep over various learning rates and track the performance. We find that for all tested individual learning rates, *A-MoD* outperforms MoD, see Fig. 13.

### 4.3 FASTER CONVERGENCE WITH *A-MoD* ON TRANSFER LEARNING

We now investigate *A-MoD* for transfer learning tasks from ImageNet-1k to three smaller image classification datasets: OxfordIIT-Pets, Stanford Cars and Flower102. These tasks pose a challenge as the pretrained model must adapt to a MoD architecture with reduced capacity while training on limited data. Fig. 4 reports the accuracy curves for *A-MoD* in comparison with MoD on Flower102 datasets. Results for Stanford Cars and OxfordIIIT-Pets datasets are provided in Fig. 10 in the Appendix.

Across all datasets and model architectures, we find that *A-MoD* converges faster in comparison to standard routing while outperforming standard routing in most cases. We analyze convergence by

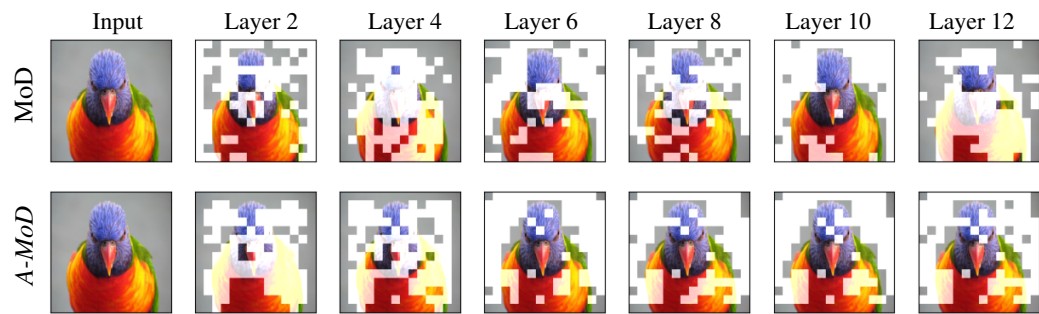

Figure 5: **A-MoD exhibits more meaningful routing compared to MoD.** Routing visualization: Example of DeiT-Small with $50\%$ capacity on ImageNet. Each example shows tokens chosen by standard MoD (top) and *A-MoD* (bottom) for every MoD layer, white patches denote skipped. Each column represents a MoD layer as depth increases from left to right.

measuring the number of epochs required for either model to reach within $2\%$ of its peak accuracy. The black dotted lines in Fig. 4 and Fig. 10 help visualize this convergence for both *A-MoD* and standard routing. For ViT-Large on Flowers, (see Fig. 4(d)), *A-MoD* reaches $94.5\%$ accuracy in the 35-th epochs, while standard routing requires 100-th epochs to reach the same value. Similarly, in case of DeiT-S on Pets, (see Fig. 10(f) in the Appendix), *A-MoD* reaches $90\%$ accuracy, in 25 epochs while standard routing takes 70 epochs to reach the same accuracy, enabling a $\sim 2\times$ speed up. These observations are consistent across architectures and datasets and highlight the effectiveness of *A-MoD* to transfer MoD models from pretrained checkpoints.

**isoFLOP comparison** We also compare *A-MoD* and standard routing to isoFLOP models on transfer learning tasks for both $50\%$ and $12.5\%$ capacity in Tables 4 and 5 in the appendix. We find that MoD models are unable to match the isoFLOP model performance on transfer tasks. We observe this as a limitation of the MoD framework in general for transfer learning on image tasks, irrespective of the routing mechanism employed. We propose a potential remedy to also outperform isoFLOP models in Section 4.5.

### 4.4 ATTENTION ROUTING IDENTIFIES IMPORTANT TOKENS

To understand why *A-MoD* improves over standard routing, we investigate the routing scores and their correlation with leave-one-out (Hastie et al., 2009) token importance. Our goal is to estimate the relationship between the importance of a token and the routing score assigned to it by a standard or *A-MoD* router. Based on our empirical results, we conjecture that *A-MoD* weights are better correlated with token importance in comparison with standard routing, thus enabling *A-MoD* to always choose the most relevant tokens.

We first verify this claim by visualizing the routing in case of individual examples from ImageNet-1k as shown in Fig. 5. The figure highlights which patches of the image are chosen by the router in each MoD layer. In case of *A-MoD* (bottom), the router selects tokens that are part of the bird outline and face starting from the third MoD layer. In contrast, standard routing (top) selects more tokens that are part of the background, up to the last layer.

Visualizing the attention maps of the last layer in Fig. 6 also confirms that *A-MoD* is able to focus on the object in the image, which we use as routing scores. The attention map for each head in the last layer for DeiT-Small identifies the silhouette of the bird for *A-MoD*, but struggles for standard routing. However, as shown by Darcet et al. (2024), attention maps do not always learn semantically meaningful scores. This holds especially for larger models, where the attention scores tend to concentrate on a single patch (token) (see Fig. 15 in the Appendix).

To quantify our qualitative observations, we compute the correlation of the routing scores with token importance estimates. For the importance of a token, we compute the change in loss of the model if that token is omitted in the vanilla transformer i.e. leave-one-out token importance. A large change in loss implies higher token importance and we would expect that token to have a higher routing

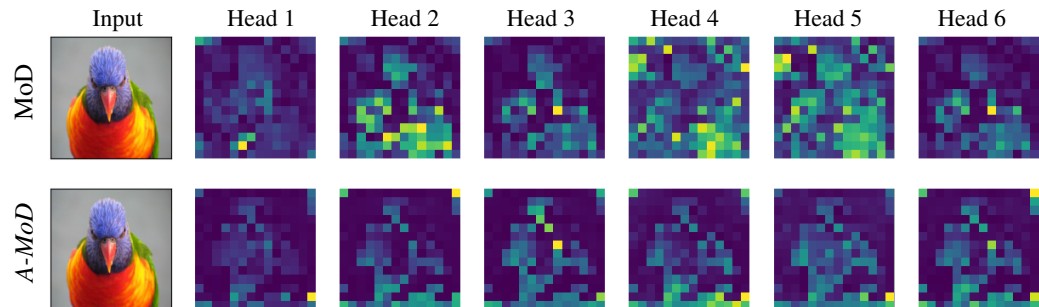

Figure 6: **A-MoD generates more meaningful attention maps compared to MoD.** Attention visualization: Example of DeiT-Small with $50\%$ capacity on ImageNet. The attention maps of the last MoD layer for standard routing (top row) and *A-MoD* (bottom row) for each example. Each column denotes an attention head of the last layer.

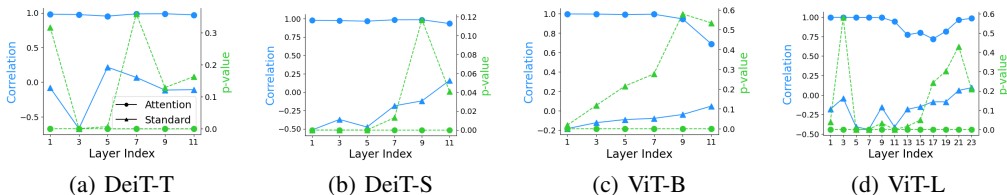

| (a) DeiT-T | (b) DeiT-S | (c) ViT-B | (d) ViT-L |
|---|---|---|---|

Figure 7: **A-MoD shows higher correlation between routing scores and leave-one-out token importance.** Correlation and p-values of the routing scores with layer-wise leave-one-out token importance on ImageNet.

score. The correlation of the routing scores for both standard routing and *A-MoD* with the token importance is shown in Fig. 7 along with the corresponding p-values.

We observe that routing scores computed by *A-MoD* consistently have a very high correlation with token importance suggesting that attention routing assigns higher scores to important tokens. In contrast, standard routing sometimes even has a negative correlation with token importance, implying that it can assign higher scores to less important tokens. Moreover, all the p-values observed for *A-MoD* were lower than $10^{-8}$, whereas they were significant (in some layers even larger than $0.5$) in case of standard routing, implying higher uncertainty in case of standard routing.

### 4.5 IMPACT OF MOD IN DIFFERENT LAYERS

In our experiments so far, MoD layers are used in alternate layers following Raposo et al. (2024). This model architecture gives us Pareto-optimal results on ImageNet-1k (see Table 1). We conduct an ablation study to investigate whether introducing MoDs only in the later layers and keeping the initial layers dense is advantageous, particularly for visual tasks like classification, where learning low-level features may be critical. In order to verify if MoDs benefit from additional feature learning at full capacity in the earlier layers, we introduce MoD layers alternately starting from the 4-th layer, keeping the first four layers dense.

Results in Fig. 8 show that keeping the first four layers dense improves on DeiT-Small and ViT-Base models trained on the Stanford Cars dataset. The additional FLOPs allows for better learning in this regime as shown in Fig. 8. With this modification, *A-MoD* is able to match the corresponding isoFLOP baseline, even for transfer learning tasks. This highlights a potential method to address the limitations of *A-MoD* mentioned in Section 4.3 at the cost of additional FLOPs.

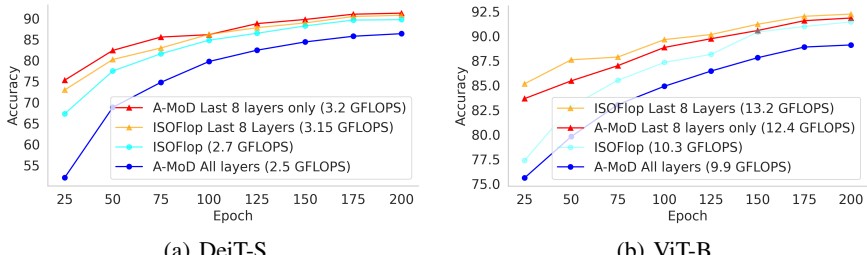

(a) DeiT-S           (b) ViT-B

Figure 8: *A-MoD* **improves the performance when only used in deeper layers.** Introducing MoDs only in the last 8 layers matches isoFLOP performance on the Stanford Cars dataset.

## 5 CONCLUSION

We propose *A-MoD*, a variation of Mixture-of-Depths (MoD) with attention routing instead of a standard router. To compute token importance for an MoD layer, *A-MoD* utilizes the attention maps from its previous layer, thereby achieving attention routing without additional parameters. In case of training from a pretrained checkpoint, leveraging trained attention information also leads to increased training stability and faster convergence compared to vanilla MoD. Furthermore, we empirically demonstrate that *A-MoD* outperforms standard MoD across different model configurations and datasets while making better routing decisions.

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

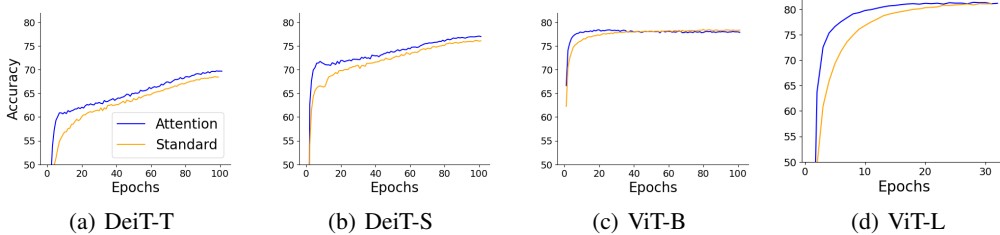

(a) DeiT-T  (b) DeiT-S  (c) ViT-B  (d) ViT-L

Figure 9: **A-MoD achieves better performance and faster convergence on ImageNet-1k.** Fine-tuning with *A-MoD*: Results comparing *A-MoD* with standard routing and isoFLOP baselines for 12.5% capacity on ImageNet-1k.

# A  APPENDIX

## A.1  MODEL SPECS

We choose four different transformer-based architectures:

## A.2  DATASETS

We evaluate our models on standard benchmark datasets: ImageNet-1K, Stanford Cars, Oxford Pets and Flowers. We use the standard train and test splits for each dataset.

- **ImageNet-1k**: A large-scale dataset comprising over 1.2 million images across 1,000 diverse categories, widely used for benchmarking image classification models (Russakovsky et al., 2015).
- **Stanford Cars**: Consists of 16,185 high-resolution images of 196 car categories, focusing on fine-grained classification tasks (Krause et al., 2013).
- **Pets**: Contains 37 categories of Pets with approximately 200 images per class (Parkhi et al., 2012).
- **Flowers**: Consists of 8,189 images of 102 flower categories, focusing on fine-grained classification tasks (Nilsback & Zisserman, 2008).

## A.3  ADDITIONAL RESULTS FOR FINETUNING ON IMAGENET-1K

Fig. 9 presents the convergence results for finetuning on ImageNet-1k with *A-MoD* at 12.5% capacity. Table 3 denotes the accuracy of *A-MoD* and standard routing without any training, after adapting the MoD weights from a vanilla pretrained checkpoint.

## A.4  COMPARISON OF *A-MoD* WITH STANDARD ROUTING AND ISOFLOP BASELINES FOR TRANSFER LEARNING

Table 4 and Table 5 present the classification accuracy of each model configuration across the three datasets for 12.5% and 50% capacity respectively. Fig. 10 presents the convergence results for *A-MoD* on transfer learnign tasks on the Stanford Cars and OxfordIIT-Pets datasets.

Table 2: Specifications of transformer-based models used in experiments

| Model | Parameters (M) | FLOPS (G) |
|---|---|---|
| DeiT-Tiny | 5.72 | 1.26 |
| DeiT-Small | 22.05 | 4.61 |
| ViT-Base | 86.57 | 17.58 |
| ViT-Large | 304.72 | 191.21 |

Table 3: **A-MoD improves adaptation.** Accuracy of MoD on ImageNet-1k, adapted from a pre-trained checkpoint, without any training.

| Model | Configuration | C = 50% | C = 12.5% |
|---|---|---|---|
| DeiT-Tiny | MoD | 4.45 | 0.42 |
|  | A-MoD | **52.6** | **0.97** |
| DeiT-Small | MoD | 0.23 | 0.16 |
|  | A-MoD | **13.49** | **0.35** |
| ViT-Base | MoD | 69.91 | 62.25 |
|  | A-MoD | **78.88** | **66.62** |
| ViT-Large | MoD | 0.43 | 0.2 |
|  | A-MoD | **49.06** | **6.03** |

Table 4: Top-1 accuracy for MoD models with 12.5% capacity, compared with isoFLOP baselines.

| Model | FLOPS (G) | Configuration | Cars | Flowers | Pets |
|---|---|---|---|---|---|
| DeiT-Tiny | 0.746 | isoFLOP | **85.09** | **90.73** | **85.3** |
|  | 0.709 | MoD | 75.77 | 81.39 | 84.05 |
|  | 0.709 | A-MoD | 78.32 | 82.82 | 84.68 |
| DeiT-Small | 2.333 | isoFLOP | **89.88** | **90.97** | 86.42 |
|  | 2.592 | MoD | 85.36 | 88.46 | 86.04 |
|  | 2.591 | A-MoD | 86.39 | 89.44 | **89.58** |
| ViT-Base | 8.849 | isoFLOP | **91.57** | 92.29 | 88.98 |
|  | 9.876 | MoD | 89.8 | **92.87** | **92.85** |
|  | 9.875 | A-MoD | 89.26 | 91.77 | 92.61 |
| ViT-Large | 33.439 | isoFLOP | **92.97** | **97.7** | **92.3** |
|  | 34.523 | MoD | 90.87 | 95.85 | 89.1 |
|  | 34.52 | A-MoD | 91.39 | 96.66 | 88.4 |

## A.5 EFFECT OF MULTIPLYING ROUTING WEIGHTS

We compare *A-MoD* with a modified version which multiplies the attention routing scores to the output of the MoD layer. Results in Fig. 11 show that multiplying the routing scores to the output can slow down convergence.

## A.6 EFFECT OF LEARNING RATES

We identify the optimal learning rates for finetuning by conducting a sweep across a range of learning rates for finetuning on ImageNet-1k as shown in Fig. 12 and Fig. 13.

## A.7 ADDITIONAL VISUALIZATIONS FOR *A-MoD*

In Fig. 14 and Fig. 15 we show the routed patches of each MoD layer and the attention maps of the last MoD layer in a ViT-Base trained on ImageNet respectively.

Additionally, we visualize the routed patches and the routing weights for a DeiT-Tiny model trained on Stanford Cars in Fig. 16.

## A.8 *A-MoD* WITH FLASH ATTENTION

Flash Attention has been proposed by Dao et al. (2022) as a method to implement attention without the need to compute the $N \times N$ attention map explicitly, reducing hardward communication overhead and thus speeding up computation considerably. Flash Attention directly computes the final output of attention $A_h^l V_h^l$ for each head based on intermediate tiling steps which compute the attention

Table 5: Top-1 accuracy for MoD models with 50% capacity, compared with isoFLOP baselines.

| Model | FLOPS (G) | Configuration | Cars | Flowers | Pets |
|-------|-----------|---------------|------|---------|------|
| DeiT-Tiny | 0.951 | isoFLOP | **88.22** | **93.42** | 87.84 |
| | 0.927 | MoD | 83.98 | 87.82 | 86.94 |
| | 0.927 | *A-MoD* | 86.9 | 89.75 | **87.92** |
| DeiT-Small | 3.47 | isoFLOP | **91.97** | **94.6** | 87.84 |
| | 3.42 | MoD | 89.58 | 91 | 91.36 |
| | 3.42 | *A-MoD* | 90.10 | 91.8 | **92.12** |
| ViT-Base | 13.216 | isoFLOP | **92.61** | **96** | 92.64 |
| | 13.105 | MoD | 91.18 | 93.62 | 92.72 |
| | 13.104 | *A-MoD* | 91.15 | 93.08 | **93.21** |
| ViT-Large | 46.241 | isoFLOP | **93.17** | 96.13 | **92.8** |
| | 45.927 | MoD | 92 | 95.69 | 89.67 |
| | 45.925 | *A-MoD* | 91.95 | **96.56** | 90.37 |

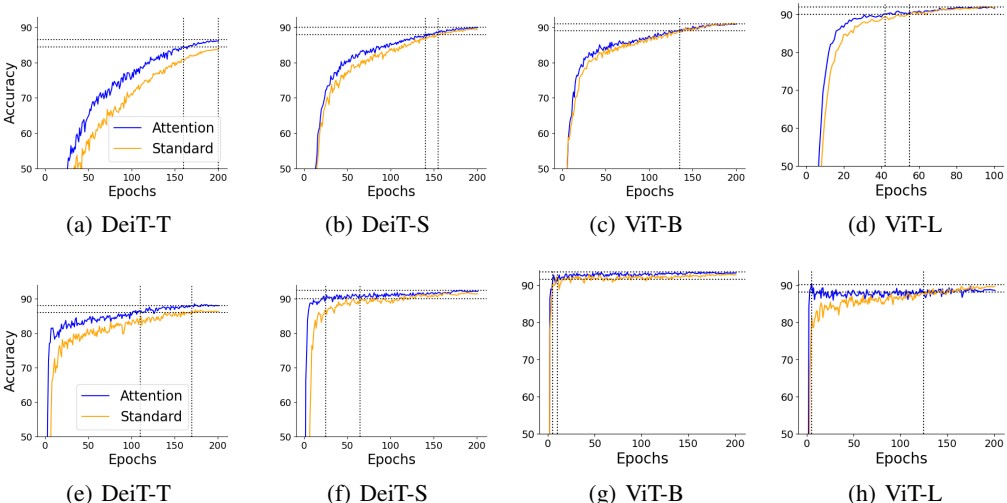

(a) DeiT-T  (b) DeiT-S  (c) ViT-B  (d) ViT-L

(e) DeiT-T  (f) DeiT-S  (g) ViT-B  (h) ViT-L

Figure 10: ***A-MoD* converges faster across different datasets** Transfer learning with *A-MoD*: *A-MoD* with 50% capacity MoD trained on the Stanford Cars (top row) and OxfordIIT-Pets (bottom row) datasets. Dotted lines denote the epochs needed to reach within 2% of peak accuracy.

map implicitly. We propose an alternate method to perform attention routing with *A-MoD* in Flash Attention framework as we do not have access to the attention map in this case. This allows efficient implementation of our method.

Referring to Algorithm 1 in Dao et al. (2022), our goal is to aggregate the attention scores from each tile $B$ and gather them in a vector of size $R \in \mathbb{R}^{N \times 1}$ which are the routing weights. Thus, we can perform attention routing with *A-MoD* without explicitly forming the $N \times N$ attention map which can be expensive in terms of computation.

Following similar notations from Algorithm 1 in Dao et al. (2022), we perform *A-MoD* as shown in Algorithm 1. Basically, for each query $Q_i$, $A_{\text{temp}}$ will be used to aggregate nonnormalized attention scores for individual tokens in line 3-7. The scores will then be normalized and accumulate to the final routing weights R in line $8 - 9$. It must be noted that we swap the order of tiling from row first (as in Dao et al. (2022)) to column first in order to aggregate row wise scores efficiently. Thus in each iteration only the $Q$ can be cached while the $K$ and $V$ need to be loaded for each tile. Our algorithm introduces a small memory overhead of $O(N)$ due to additional temporary variables.

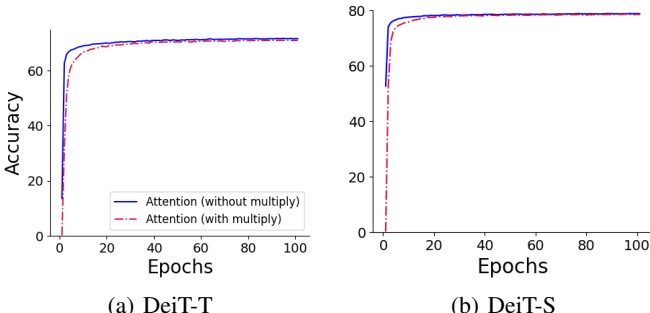

(a) DeiT-T          (b) DeiT-S

Figure 11: **A-MoD with routing scores multiplied to MoD output**. Multiplying the output of the MoD block with routing scores for *A-MoD* (red curve) compared to the proposed *A-MoD* without multiplication (blue curve).

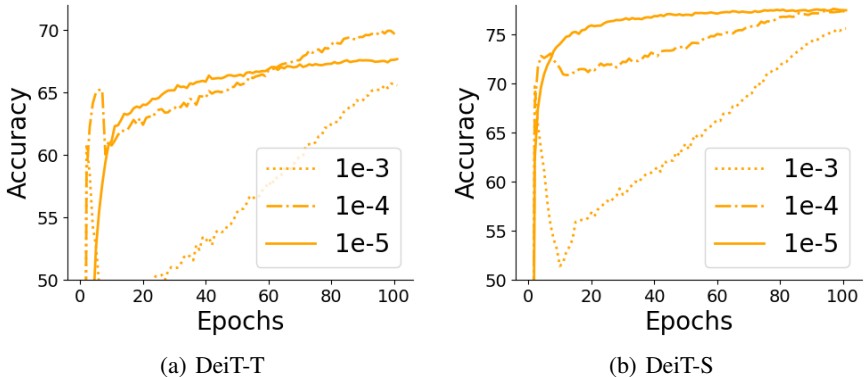

(a) DeiT-T          (b) DeiT-S

Figure 12: Sweep over learning rates on ImageNet-1k for standard routing.

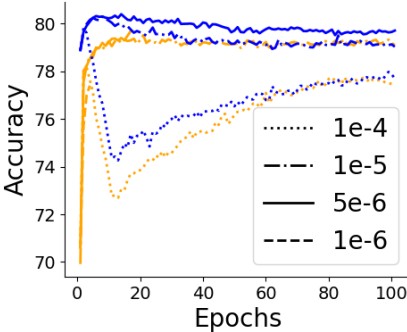

Figure 13: Sweep over learning rates on ImageNet-1k for standard routing and *A-MoD* on ViT-Base. Orange curves denote standard routing and blue curves denote attention routing.

### A.9 COMPARISON WITH TOKEN PRUNING METHODS

We compare our method *A-MoD* with other token-pruning and token-merging including Token Mergin (ToME) (Bolya et al.), A-ViT (Yin et al., 2022) and Dynamic-ViT (Rao et al., 2021) to validate the performance of *A-MoD*. We compare with the baseline results provided in Table 11 in Bolya et al. and Table 3 in Yin et al. (2022). However, we note that ToMe (Bolya et al.) trains their models with distillation while the other methods do not, which aids ToMe. Results are provided in Table 6 and Fig. 17

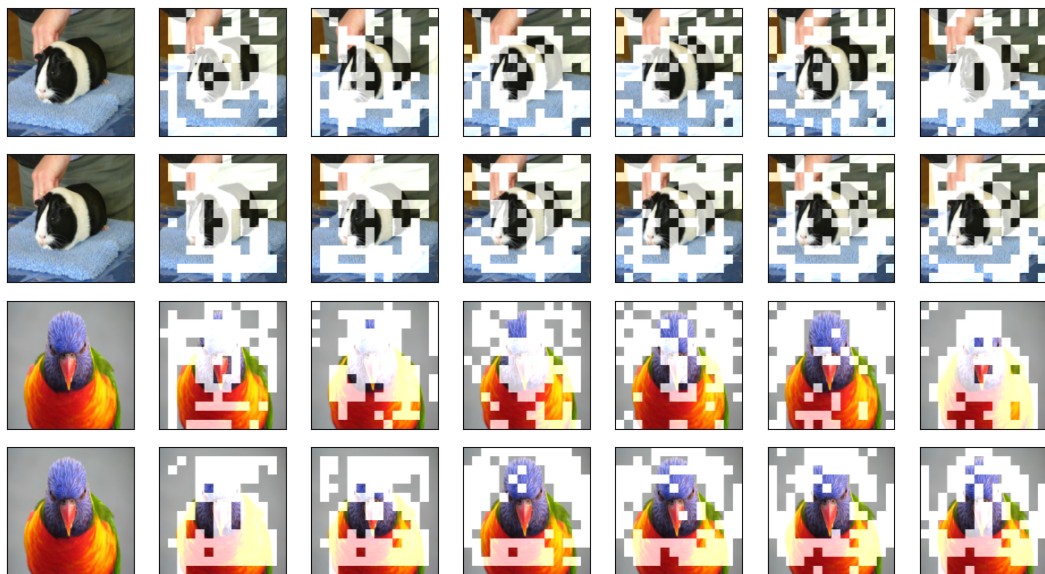

Figure 14: Routing example on ViT Base with 50% capacity trained on ImageNet-1k. Each example shows tokens chosen by standard routing (top) and attention routing (bottom). Each column represents a MoD layer as depth increases from left to right.

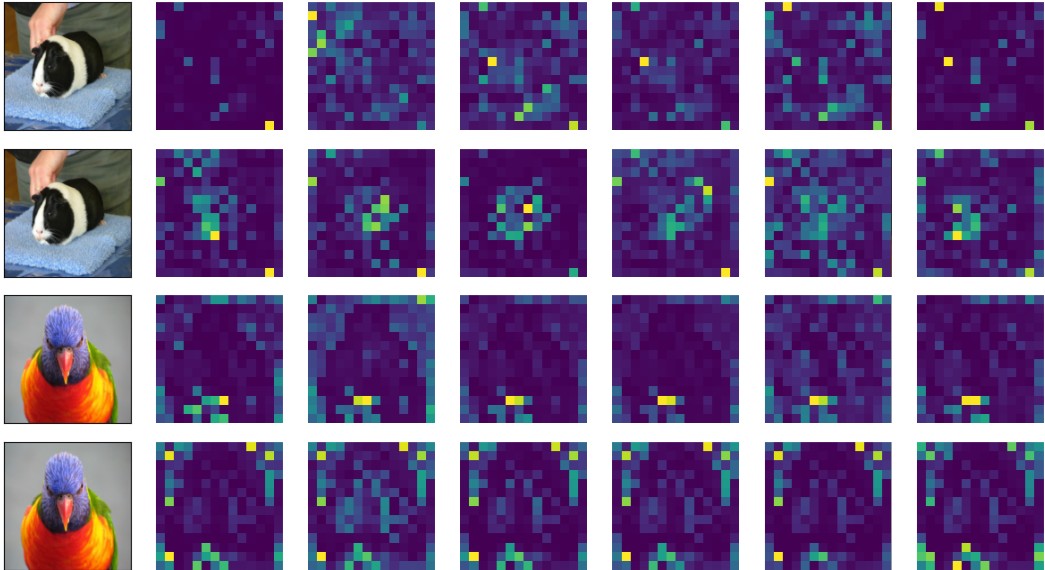

Figure 15: Attention map of each head in the last layer of a ViT Base MoD with 50% capacity for standard routing (top) and attention routing (bottom) finetuned on ImageNet-1k. Each column denotes an attention head of the last layer.

## A.10 MODEL THROUGHPUT

We provide a comparison of model throughput in Fig. 18. *A-MoD* has a higher throughput (img/s) in comparison to MoD and isoFLOP baselines. We also provide a breakdown of each method using the PyTorch profiler to highlight the CPU and GPU time used by each method for both the Attention layer and the MLP layer as shown in Fig. 19.

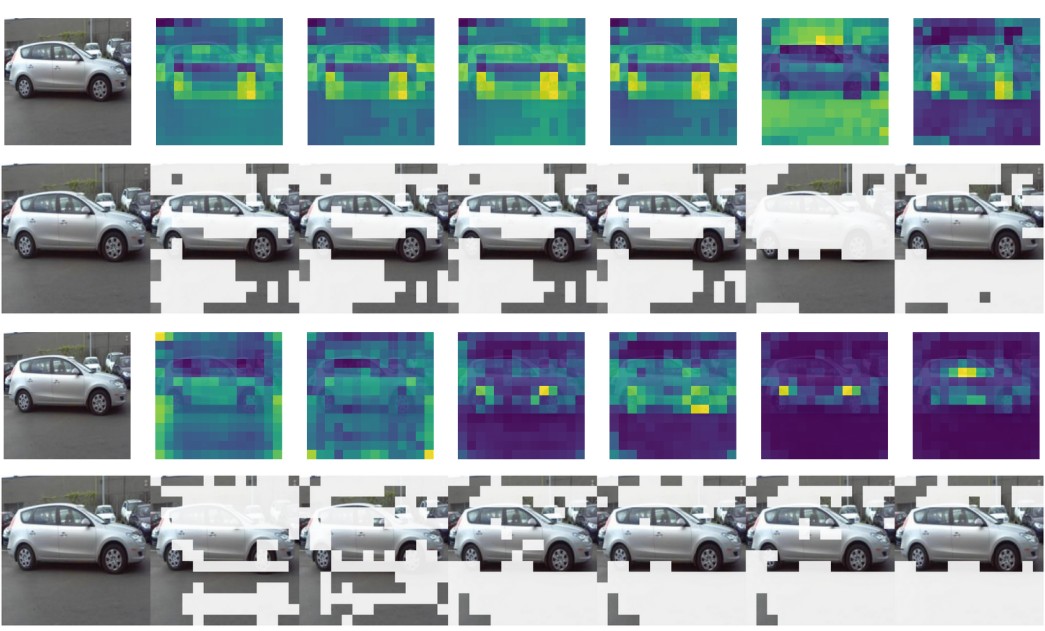

Figure 16: Visualizing routing scores and token selection of each MoD layer of a DeiT-Tiny MoD with 50% capacity for standard routing (top) and attention routing (bottom) finetuned on Stanford Cars.

---

**Algorithm 1** *A-MoD* with Flash Attention, modified from Algorithm 1 in Dao et al. (2022)

---

1: Initialize, $R \in \mathbb{R}^{N \times 1}$ and $A_{\text{temp}} \in \mathbb{R}^{B_r \times N}$ to zero.
2: **for** each $i$ from 1 to $T_r$ **do**
3:     **for** each $j$ from 1 to $T_c$ **do**
4:         $S_{ij} = Q_i K_j^T$
5:         $P_{ij} = \exp(S_{ij})$
6:         Accumulate for row sum as, $A_{\text{temp},j} = P_{ij}$
7:     **end for**
8:     $l_i = \sum_j A_{\text{temp},j}, l_i \in \mathbb{R}^{B_r}$
9:     $R \leftarrow R + P_{ij}/l_i$
10: **end for**
11: **return** Routing weights R

---

Table 6: Comparison with other token-pruning and merging methods (* denotes training with distillation).

| Model | Method | Top-1 Acc (%) | FLOPs (G) |
|---|---|---|---|
| **DeiT-T** | A-MoD | 71.8 | 0.9 |
| | A-ViT | 71.0 | 0.8 |
| | Dynamic ViT | 70.9 | 0.9 |
| | ToMe (with distillation) | 71.69* | 0.93 |
| **DeiT-S** | A-MoD | 78.66 | 3.42 |
| | A-ViT | 78.6 | 3.6 |
| | Dynamic ViT | 78.3 | 3.4 |
| | ToMe (with distillation) | 79.68* | 3.43 |

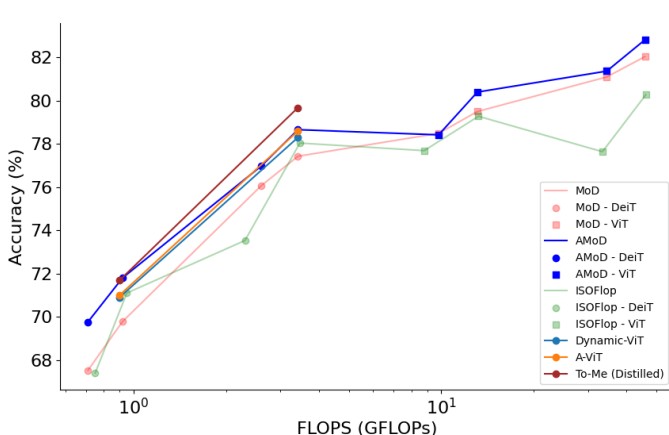

Figure 17: Accuracy Comparison with token-pruning and merging methods

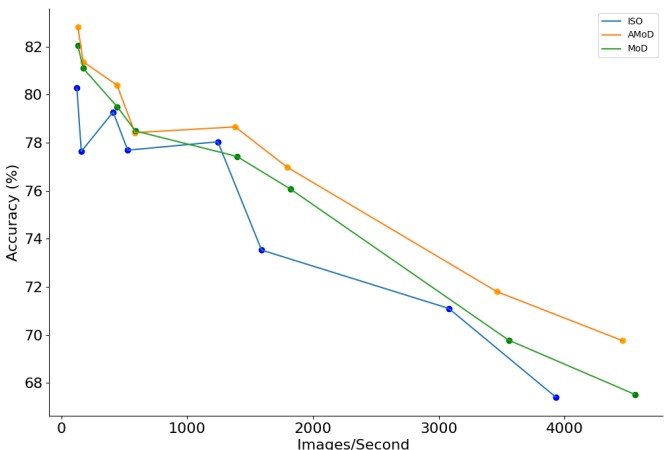

Figure 18: Accuracy vs Throughput for MoD vs ISOFlop Models with Batch Size 100 on Nvidia A100 GPU.

## A.11 EFFICIENCY OF *A-MoD*

To highlight the efficiency of A-MoD, we compare it with the baseline DeiT-S and report the top-1 accuracy on ImageNet. A-MoD is able to reduce the number of FLOPs by up to $18\%$ without dropping performance, with standard training and no additional tricks. Results are provided in Table 7.

Table 7: Comparison of A-MoD ($70\%$ capacity) with the vanilla DeiT-S baseline which has more FLOPs.

| Model | FLOPs (G) | Top-1 Accuracy (%) |
|---|---|---|
| DeiT-S Baseline | 4.6 | 79.6 |
| A-MoD (C = $70\%$) | 3.8 | 79.63 |

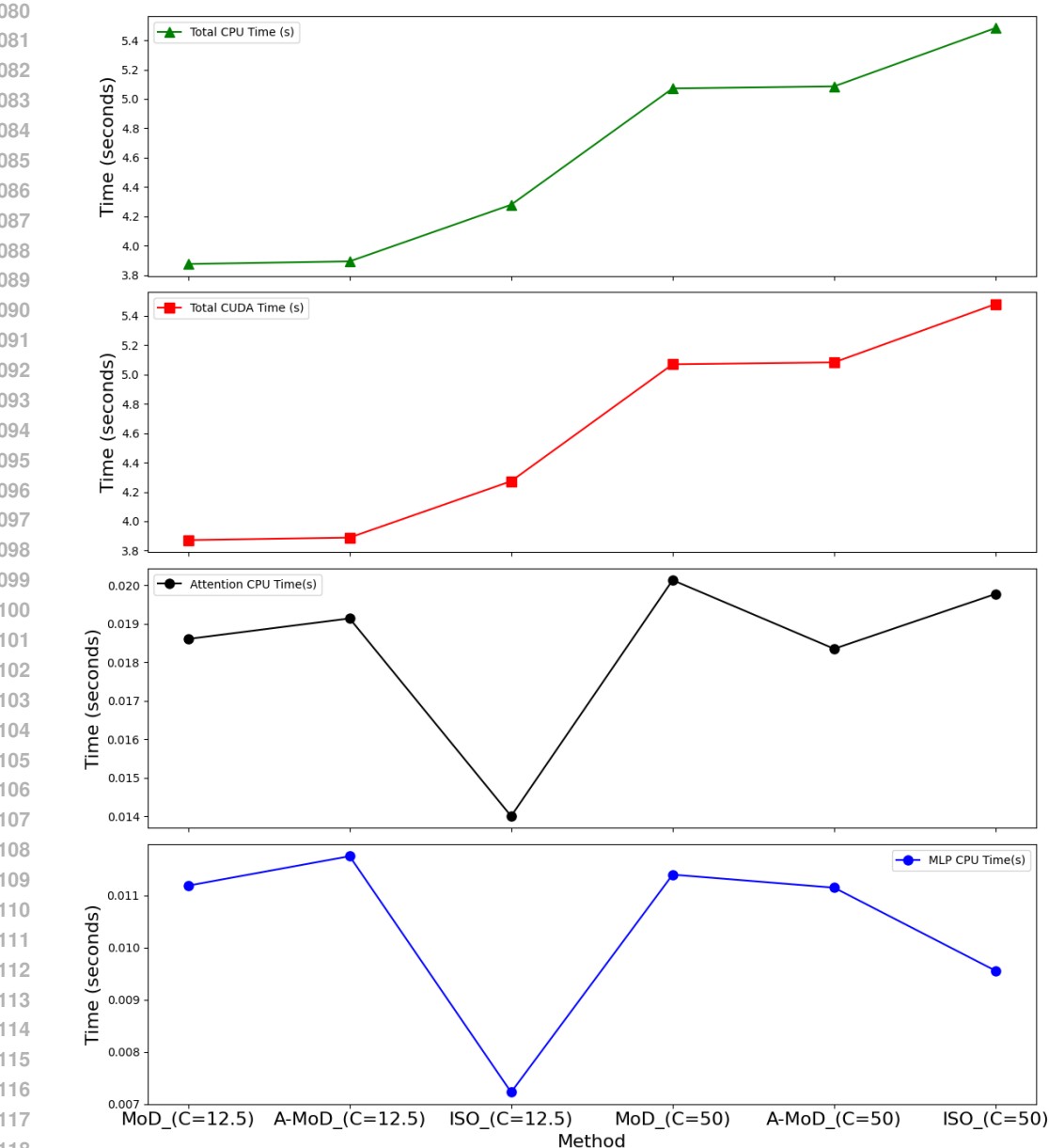

Figure 19: Profiling *A-MoD*, MoD and isoFLOP **ViT-Base** methods on Nvidia A100 GPU. The x-axis shows different models from left to right: MoD, *A-MoD* and isoFLOP for both C=12.5% and C=50%.

## A.12 TRAINING FROM SCRATCH

We also provide results for training from scratch on ImageNet and observe that A-MoD outperforms standard routing as shown in Table 8.

## A.13 RESULTS ON OBJECT DETECTION

We also provide results with the DETR architecture (Carion et al., 2020b) for *A-MoD*. Table 9 shows that MoD and *A-MoD* achieve comparable results in this case.

Table 8: Training from scratch comparison for *A-MoD* and MoD on ImageNet-1k.

| Model | Training Epochs | Method | Accuracy (%) |
|-------|-----------------|--------|--------------|
| DeiT-S | 300 | A-MoD | 76.63 |
|        |     | MoD | 75.90 |
| ViT-Base | 160 | A-MoD | 73.66 |
|          |     | MoD | 72.47 |

Table 9: Comparison of mAP and FLOPs across DETR MOD, DETR *A-MoD*, and DETR-Baseline.

| Model | MoD, C=50% | *A-MoD*, C=50% | Baseline |
|-------|-----------|----------------|----------|
| **mAP** | 39.6% | 38.6% | 39.9% |
| **FLOPs(G) (Total)** | 83.2 | 83.2 | 86.56 |
| **FLOPs(G) (Transformer Encoder/Decoder)** | 7.747 | 7.745 | 10.745 |

