# OpenReview forum: "Attention Is All You Need For Mixture-of-Depths Routing"
_ICLR.cc/2025/Conference — Submitted to ICLR 2025_

### Official Review · Reviewer_je9B · 2024-10-27

**Soundness:** 2
**Presentation:** 3
**Contribution:** 2
**Rating:** 5
**Confidence:** 4

**Summary:**

Mixture of Depth methods have recently been proposed as an alternative to Mixture of Expert methods. Instead of deciding to which expert to route a token, MoD takes a decision whether or not to skip the current expert. The main purpose shared by these methods is that of finding a better compute/performance tradeoff, which is currently of particular interest in the context of LLMs and large generative models. One of the main challenges for building such methods is the design of stable routing mechanisms. The authors propose A-MoD, a routing method which performs pruning of tokens traversing the current layer, based on the respective attention scores in the previous layer. The authors evaluate their method on classification tasks on a set of DeiT and DiT models of varying size.

**Strengths:**

- The method is simple and clearly explained throughout the paper. It seems possible to implement it easily for small models operating on short token sequences.
- The proposed method is simple and the idea of eliminating learnable routing, a major source of training instabilities, is an interesting research direction.
- Some performance improvements are shown under some settings of a classification task.

**Weaknesses:**

- The method requires access to the attention matrices. Modern attention implementations (e.g. "FlashAttention") do not materialize attention matrices due to their large materialization cost. An implementation of A-MoD viable for an LLM or large generative model would require non-trivial implementation effort to match the performance of modern attention implementations.
- The chosen evaluation setting is composed only of classification tasks. Token pruning is particularly suited for this task, as all tokens with information not relevant to the depicted object can be safely discarded and the model output consists in only an aggregation of the tokens. A more significant evaluation would have taken segmentation, language modeling, or image/video generation problems into account. In these tasks token pruning can be problematic, as every token will be returned as an output. Pruned tokens are at risk of generating artifacts in such context. Lack of evaluation in these context is a significant limitation.
- The method presents a limitation in the maximum achievable speedup with respect to a corresponding dense architecture. In particular, as each A-MoD block bases its pruning on the previous block, it can only be applied at alternating blocks, posing a maximum achievable speedup to 2X over a corresponding non A-MoD model. In contrast, a learnable routing mechanism would allow application of MoD to each layer, allowing for higher theoretical speedups, making the pursuit of this paradigm more promising than A-MoD.
- Some recent MoE architectures (Snowflake Arctic LLM) show that it is beneficial to apply MoE at every layer. A-MoD can apply it only every other layer
- Table 1 proposed evaluation with an isoFLOP baseline. Under 3/8 configurations, the isoFLOP baseline has <90% of the flops of the A-MoD model. Such comparison does not appear fair
- Table 1 would benefit from being accompanied by a Pareto curve visualization of performance vs compute, which could highlight better the Pareto front for the C=12.5% case, where compute of the baseline and A-MoD method is mismatched.
- The paper does not take inference throughput into consideration. In this paper, FLOPs are an incomplete indicator of model cost in that they do not highlight performance losses caused by the need to materialize the attention matrix. It is likely that the isoFLOP baseline when implemented carefully with an efficient attention implementation would outperform A-MoD in terms of inference throughput and memory utilization, especially if evaluated in practically-relevant language modeling or generative tasks with large amounts of tokens.
- Discussion in 4.5 and Fig. 7 are not convincing. In Fig 7 (a) performance is aligned to the isoFLOP baseline, in Fig 7 (b) compute is too mismatched between baselines to draw conclusions.

In summary, the pursuit of A-MoD seems less promising than traditional MoD or MoE paradigms due to its limited theoretical maximum achievable speedup of 2X. Evaluation is performed on classification only, missing some important performance metrics (throughput/latency) and in some circumstances under unfair evaluation settings for the baseline. Evaluation does not comprise important dense prediction tasks such as language modeling or image/video generation. No discussion or remedy is presented for the need of materializing full attention matrices, limiting applicability of the method for large models operating on long sequences, which could benefit most from MoD/MoE.

For these reasons, I do not believe the method implementation and evaluation are solid enough for the paper to be accepted.

**Questions:**

- Can the authors provide an implementation of an efficient attention mechanism such as "Flash Attention" with support for A-MoD? If yes, how would the performance of this implementation compare to the original implementation?
- Can the authors show performance of the method on a task such as semantic segmentation, language modeling or image/video generation that involves producing an output token for each input token to confirm that token dropping would not affect output quality?
- Could the authors revise Table 1 to perform a fairer comparison with isoFLOP baselines and show results using a Pareto curve?
- Could the authors revise qualitative evaluations to include throughput measures rather than FLOPS only, ensuring baseline methods make use of efficient attention implementations?
- Could the authors revise quantitative evaluation to include a task involving large numbers of tokens (>=4k tokens) to show that the attention operation implementation can remain efficient as the input size grows?

---

> ### Author Response · Authors · 2024-11-22
> **Reply to Reviewer je9B**
>
> We thank the reviewer for the detailed feedback.
>
> As suggested by the reviewer, our method A-MoD is a simple routing mechanism that can be easily integrated in dynamic computation models, specifically into MoD which is the focus of our work. Our main contribution is to highlight that there is no need for a dedicated router in MoD models, which needs additional tuning and can lead to training instability. The information required to perform routing is already present in the network which we leverage from attention maps, via A-MoD.
>
> 1. **Integration to Flash Attention**: A-MoD is in fact directly compatible with Flash Attention without having to compute the NxN matrix explicitly. Token importance score can be aggregated iteratively together with Flash Attention tiling strategy. This requires additional O(N) memory overhead for intermediate variables. We provide an algorithm for the same in Appendix A.8 of our updated draft.
>
> 2. **Non-classification tasks**: We also provide additional experiments on object detection with A-MoD as suggested by the reviewer (please see our general response and Table 9 in the updated draft).
>
> 3. **Maximum achievable speedup**: In our experiments for  MoD with standard routing, we find that placing MoDs in each layer does not improve the performance and often leads to unstable training. Intuitively, in earlier layers it is often not yet clear, which tokens are relevant, and which can be skipped [6]. Thus, achieving the theoretical larger speedups is empirically difficult. Note that a denser placement of A-MoD layers, especially in later layers, could be achieved by re-using prior attention matrices (e.g., from 2 layers before) and relying on the fact that embeddings change slowly [5].
> Nonetheless, as shown in our comparisons with SOTA baselines, A-MoD is competitive and even better than other SOTA token merging and token pruning methods like Dynamic ViT[3] and A-ViT [4] and ToMe[2] (see our general response for results).
>
> 4. **Closer IsoFLOPs**: We design our isoFLOP models by reducing the depth of the vanilla ViT architectures such that our isoFLOP models can also benefit from the pretrained weights of the baseline models.
>   - We would like to clarify that, in majority of the cases (5/8), the isoFLOP models have a greater number of FLOPs than the A-MoD models, and yet, A-MoD is able to outperform them as shown in Table 1. We agree, in (2/8) cases, the number of FLOPs for the isoFLOP model <90%. Please refer to Figure 16 in the Appendix for a Pareto-curve of accuracy vs FLOPs.
>   - Additionally,  we show an exact comparison between A-MoD and the isoFLOP baseline on DeiT-Small for 2.6GFlops by reducing the embedding dimension of the isoFLOP model to get more finegrained control over the FLOPs.
>       - A-MoD (12.5% Capacity, 2.6GFlops): 76.98%
>       - MoD (12.5% Capacity, 2.6GFlops): 76.07%
>       - IsoFLOP (2.6GFlops): 75.73%
>
> 5. **Pareto front visualization and throughput**: Please see general response and the updated draft for accuracy vs FLOPs (Appendix A.9, Figure 16), accuracy vs throughput (A.10, Figure 17) and run-time analysis of our method.
>
> Q1. We provide an algorithm compatible with Flash Attention in Appendix A.8.
>
> Q2. Please see our general response for results on object detection (See Table 9 in Appendix).
>
> Q3. We provide a Pareto curve in Figure 16 in the Appendix and make an exact isoFLOP comparison for DeiT-S.
>   - A-MoD (12.5% Capacity, 2.6GFlops): 76.98%
>   - MoD (12.5% Capacity, 2.6GFlops): 76.07%
>   - IsoFLOP (2.6GFlops): 75.73%
>
> Q4. Throughput results are provided in Figure 17 in the Appendix .
>
> Q5. A-MoD does not affect the scalability of Tranformer models w.r.t. the input sequence length and is compatible with efficient attention modules like Flash Attention (see Algorithm 1 in Appendix A.8). Hence, we believe that A-MoD still performs well in the regime with >4k tokens. However, as our paper focuses on the vision domain, where large sequence lengths are not typical, we leave this exploration to future work.
>
> [1] Jain, Gagan, et al. "Mixture of nested experts: Adaptive processing of visual tokens." arXiv preprint arXiv:2407.19985 (2024).
>
> [2] Bolya, Daniel, et al. "Token Merging: Your ViT But Faster." The Eleventh International Conference on Learning Representations.
>
> [3] Rao, Yongming, et al. "Dynamicvit: Efficient vision transformers with dynamic token sparsification." Advances in neural information processing systems 34 (2021): 13937-13949.
>
> [4] Yin, Hongxu, et al. "A-vit: Adaptive tokens for efficient vision transformer." Proceedings of the IEEE/CVF conference on computer vision and pattern recognition. 2022.
>
> [5] Liu, Zichang, et al. "Deja vu: Contextual sparsity for efficient llms at inference time." International Conference on Machine Learning. PMLR, 2023.
>
> [6] Puigcerver, Joan, et al. "From Sparse to Soft Mixtures of Experts." The Twelfth International Conference on Learning Representations.

---

> ### Comment · Reviewer_je9B · 2024-11-23
>
> I'd like to thank the authors for their comprehensive and concise responses.
> The following points have been addressed by their rebuttal:
> - Integration with Flash Attention
> - Fairer comparison with IsoFLOP baseline
> - Pareto curve visualizations
> - Throughput analysis
>
> My concerns remain regarding the limited scope of the study (as shared by other reviewers), though I appreciate the inclusion of initial object detection results. I'd like the method to be evaluated on more dense prediction tasks such as segmentation or image generation.
> My concern regarding the reduced attractiveness of A-MoD compared to traditional MoD or MoE paradigms remains.
> I will update my score accordingly

---

> ### Author Response · Authors · 2024-11-27
> **Reply to Reviewer je9B**
>
> We thank the reviewer for acknowledging our response and their positive comments.
>
> Regarding the additional concerns,
>
> 1. We are the first to provide vision experiments on ImageNet classification for the MoD architecture, based on which we develop A-MoD to improve token selection based on attention. Our preliminary results suggest that we are able to extend the benefits of A-MoD to other vision tasks including detection. A comprehensive investigation of segmentation and detection lies outside the scope of this work, and we would dedicate time for it as part of future work.
>
> 2. We would also like to highlight that both MoD and A-MoD suffer from trainability issues in the case where each layer is accelerated with token skipping. However, if we keep the first few layers dense (no tokens skipped), we can leverage attention maps from those layers to implement A-MoDs in each of the successive layers. Similarly, standard routers would learn better in later layers allowing better speed up for MoDs when initial layers are kept dense. Please also note that MoEs are a different paradigm that are orthogonal to MoDs and can provide additional speed-up in combination with MoDs [1].
>
> We are happy to clarify any further questions the reviewer might have and continue this discussion.
>
> [1] Raposo, David, et al. "Mixture-of-Depths: Dynamically allocating compute in transformer-based language models." arXiv preprint arXiv:2404.02258 (2024).

---

### Official Review · Reviewer_N1Y6 · 2024-11-02

**Soundness:** 2
**Presentation:** 2
**Contribution:** 2
**Rating:** 3
**Confidence:** 4

**Summary:**

In this work, the authors aims to deal with an interesting problem: How to reduce the token number need to be computed in a layer, and proposed A-MOD. A-MOD is simply based on attention score to select tokens need to be calculated, and the rest will be directly skiped. Experimental results show promising results.

Such a topic is always interesting and important, especially in the era of large models. However, almost none is applied to indutry because these methods are always only efficient in theory. I encourage more explorations in this topic. Regarding this work, I believe many experimental settings are not reasonable or not as expected.

**Strengths:**

1. The authors tackle an important issue in improving the efficiency of ViTs by proposing a method that dynamically selects the most relevant tokens for computation. This approach aligns with similar techniques in the field, such as A-ViT, which also prioritize token selection for efficiency gains.

2. Additionally, the experimental results show promising performance.

**Weaknesses:**

The main idea behind the proposed method makes sense to me, but I have several concerns, particularly regarding the experiments and their settings.

1) First, the authors mention in Line 234 that they continue training from a previous checkpoint for an additional 100 epochs. I’m curious whether this approach is justified. Why not simply start training from scratch?

2) I also find the transfer learning setup a bit confusing, especially since the authors do not use fixed pretrained weights. Why is transfer learning considered in a work that focuses on efficiency? How does it contribute to the overall goals of the work? Clarity on this point would be help.

3) Regarding the baseline, I expected to see a vanilla ViT for comparison, but instead,  the authors reduced the original ViT layers to match FLOPs directly. This choice makes no sense to me if the goal is to demonstrate that A-MOD improves effiency. Additionally, I noticed that A-MOD significantly lowers the original performance of ViT. For instance, the original ViT-Tiny achieves ~74% accuracy, whereas A-MOD only reaches 69.76% and 71.8%.
The abstract is is also very misleading; I initially was thinking the method reduces the number of tokens while improving accuracy on ImageNet by 2% (a huge improvement on ImageNet).

4) Is the same selection ratio applied across all layers, from shallow to deep? we know that shallow layers typically do not generate meaningful attention maps.

5) At a high level, A-MOD aims to reduce token number for computation, which is similar to the approach taken in the TO-ME paper (“Token Merging: Your ViT but Faster,” ICLR 2023). However, I noticed that the performance of A-MOD is much worse, and there are no speed tests provided to show efficiency gains.

6) In Fig. 4,  the last image of the first row, it appears that MOD selects almost all unimportant boundary patches and parts of the bird's head. Does this make sense? If it does, I would like to know why.

7) Generally, attention maps in shallow layers do not reliably indicate importance. Given this, how should we interpret the results shown in Fig. 5? Were the specific attention heads or examples chosen selectively, or could you provide additional much more examples (besides the bird and car used throughtout the paper) to support the findings?

8) Finally, in Line 420, the authors state, “attention maps do not always learn semantically meaningful scores.” This leads me to two questions: How did you conclude that this issue is only limited to larger models? If this only happens in larger models, why does A-MOD work for ViT-Tiny?

**Questions:**

1. Are there any other reproducible results for MOD on ImageNet?

2. How are different heads within a single layer handled? Will different heads select different tokens for computation in a layer?

3. As indicated in the abstract, it’s unclear how this method improves training and inference efficiency on an ImageNet-scale dataset. Are these efficiency improvements achieved in practice (like in training and inference speed) or just in theory?

---

> ### Author Response · Authors · 2024-11-22
> **Reply to Reviewer N1Y6**
>
> We thank the reviewer for taking the time and providing detailed feedback for our work.
>
> As pointed out by the reviewer our method A-MoD shows promising performance across provided experiments.
>
> **Experimental Setup: Training from a pretrained checkpoint**
>
> Our experiments are designed to showcase that A-MoD is a superior form of routing in terms of accuracy, when compared with the standard dedicated router as well as isoFLOP baselines. We build on the work of Raposo et al. [1], who have shown that MoD architectures are efficient, i.e., Pareto-optimal. We further show that the drawbacks of MoD architectures like training instability and router tuning can be easily eliminated with A-MoD.
>
> 1. We choose to start our experiments from a pretrained model to highlight that attention routing requires minimal training when adapting to a MoD layer from a standard dense architecture (see also Table 3 in the appendix). Our experimental setup starts from a model with pretrained weights and verifies the performance of A-MoD on a variety of datasets including finetuning and transfer learning.
>    - Our results show that A-MoD outperforms isoFLOP baselines on ImageNet-1k and also converges faster on transfer learning tasks.
>    - We also provide results for training from scratch:
>      - DeiT-S after 300 epochs
>        - A-MoD 76.63%
>        - MoD 75.9%
>      - ViT-Base after 160 epochs
>        - A-MoD 73.66%
>        - MoD 72.47%
>
> 2. We choose the transfer learning setting to test the adaptability of MoD models and find that A-MoD models converge faster than standard routing.
>
> **Efficiency and comparison to vanilla ViT**
>
> 3. We have updated the abstract to make our experimental setup and results clear. Our main contribution is that A-MoD is able to outperform standard routing and its isoFLOP counterparts. Additional comparisons with other SOTA baselines also show that A-MoD is comparable and even better than these methods. While the focus of our work is not on efficiency, we provide additional results on efficiency as well where A-MoD can achieve the same performance as the baseline with 18% fewer FLOPs (please also see general response and Table 7 in the Appendix):
>  - DeiT-S baseline (4.6G FLOPS): 79.6%
>  - A-MoD (3.8G FLOPS): 79.63%
>
> We also provide a Pareto-curve for accuracy vs FLOPs in Figure 16 and accuracy vs inference time in Fig 17 that highlights the pareto-optimality of A-MoD.
>
> **Layerwise capacity**
>
> 4. We apply the same capacity to each MoD layer. Further, we have ablated the placement of the MoD layers by keeping the first few layers dense to allow improved feature learning and observe an improvement in comparison with the isoFLOP baseline as shown in Figure 7.
>    - We believe that the capacity of each layer can be different in each layer to further improve A-MoD. However, finding the optimal per-layer capacity values is far from trivial, and we consider it as part of future work.
>
> **Comparison with SOTA baselines**
>
> 5. We provide a comparison with state-of-the-art token pruning/merging methods including Dynamic ViT, A-ViT and Token Merging and find that A-MoD is comparable (even better) than these methods (please see the general response for results and Table 6 in the updated draft).
>
> **Token importance from attention maps**
>
> 6. Figure 4 aims to differentiate between the tokens picked by standard routing (top row) and A-MoD (bottom row). Each column denotes a MoD layer. Standard routing struggles to identify the bird and only picks the background tokens, while A-MoD learns to pick the appropriate tokens representing the bird’s head starting from the third layer as the attention features become more meaningful in the later layers. Additional images are provided in the Appendix (see Figure 15).
>
> 7. In Figure 5, the attention maps are represented for every head in the last layer. Figure 5 highlights that A-MoD attention maps can better identify the objects in the image. Please see the Figure in our updated draft.
>
> 8. We mention that attention maps do not always learn semantic meaning (in line with the findings of [2]). However, as highlighted in Figure 6, attention scores do have a very high correlation with leave-one-out token importance in every layer validating their use as routing weights.

---

> > ### Author Response · Authors · 2024-11-22
> > **Reply to Reviewer N1Y6 continued**
> >
> > Q1. Please see our general response for additional experiments on ImageNet, comparing to several state-of-the-art methods on token pruning and merging. These experiments show that A-MoD is competitive and even better than these methods.
> >
> > Q2. As shown in Equation 4, we compute the average attention across all heads. That means that the same set of tokens is processed by all heads of the subsequent A-MoD layer.
> >
> > Q3. Our method A-MoD achieves faster training in comparison to standard routing with MoD architectures, both in the number of required training iterations, and in the wall time. Moreover, it does not require additional hyperparameter tuning of the router. Hence, A-MoD is a faster and better performing version of the standard MoD architecture. We also show improvements in inference time (please see general response and Figure 17 in the updated draft).
> >
> >  [1] Raposo, David, et al. "Mixture-of-Depths: Dynamically allocating compute in transformer-based language models." arXiv preprint arXiv:2404.02258 (2024).
> >
> > [2] Darcet, Timothée, et al. "Vision Transformers Need Registers." The Twelfth International Conference on Learning Representations.

---

> > > ### Comment · Reviewer_N1Y6 · 2024-11-25
> > > **Thanks for rebuttal**
> > >
> > > Thank you for your rebuttal. Below are some follow-up questions based on your responses to my initial questions.
> > >
> > > **Q1**
> > > In your general response, I was unable to find reproducible results for MOD on ImageNet (apologies if I missed them due to the volume of replies). Could you clarify whether such results exist? If they do, could you point them out explicitly here? Alternatively, is this the first work to explore this and provide the result?
> > >
> > > **Q2**
> > > Regarding the statement in the rebuttal: “by averaging the corresponding attention values across all rows and attention heads”—was this statement added recently? I’m also wondering if averaging attention across all heads is a reasonable solution. I believe this would ignore distinguishing features across different heads and potentially oversmooth the attention map? Personally, I think this operation questionable. Could you elaborate on why it is considered reasonable?
> > >
> > > **Q3**
> > > I have questions about Figure 17, particularly why ISOFLOP runs slower than A-MOD.
> > >
> > > For A-MOD, as illustrated in Figure 1, additional routing operations are introduced, along with computations for two branches. While reducing the token number (224/16)**2 likely doesn’t significantly boost speed (since it is not a large number), reducing the number of layers (ISOFLOPs) should noticeably increase speed.
> > >
> > > Given these observations, how is A-MOD (which is deeper and involves additional routing) running faster than ISOFLOP (which has fewer layers) under the same FLOPs?
> > >
> > > -------
> > > I will response to my concerns (Section Weaknesses) later. Meanwhile, could you please provide a copy of submission version pdf since current pdf is updated without emphasizing which parts are modified. Thanks for your rebutal.

---

> > > > ### Comment · Reviewer_N1Y6 · 2024-11-25
> > > >
> > > > Here are my response to authors' rebuttal
> > > >
> > > > (1) Thanks for provide addtional results; What about ISOFLOP variant train from scratch?  (A-MoD 76.63%, MoD 75.9%)
> > > >
> > > > (2) I still could not get the motivation why transfer learning, but that is ok.
> > > >
> > > > (3) Thanks for updating the abstract: original submission is misleading and now is better.
> > > >
> > > > (4) Thanks for make it clear.
> > > >
> > > > (6) Sorry that my question is: why "MOD selects almost all unimportant boundary patches and parts of the bird's head? Does this make sense". A detailed explanation is appreciate.
> > > >
> > > > --------
> > > >
> > > >
> > > > Addtional questions:
> > > >
> > > > (A) As shown in Table 5, ISOFLOPs achieves the best performance and, importantly, is much simpler.
> > > >
> > > > (B) In Table 3, the results for different sizes of ViT variants appear inconsistent. Could you explain the reasons behind this behavior?
> > > >
> > > > (C) For the ISOFLOP implementation, why not reduce the dimension to match the FLOPs instead? Wouldn't this approach be more reasonable, allowing the depth to remain consistent with MOD and A-MOD?

---

> > > > > ### Comment · Reviewer_N1Y6 · 2024-11-25
> > > > >
> > > > > Thanks a lot for authors' rebuttal and efforts. I read the rebuttal and checked the paper in details again, and would like to keep the score.

---

> ### Author Response · Authors · 2024-11-27
> **Reply to Reviewer N1Y6**
>
> We thank the reviewer for acknowledging our response,
>
> Q1. We are the first to introduce MoDs for vision classification tasks and benchmark it on ImageNet. Prior to our work, MoDs had only been introduced on language tasks. On vision tasks, we observe that standard MoDs suffer from routing instability and hence we propose A-MoD for improved routing. As highlighted in our Pareto-curve, A-MoD is even better that Dynamic-ViT and A-ViT (and is competetive with ToMe) which are state-of-the-art token-merging and token-pruning methods.
>
> Q2. We average across all the attention heads as highlighted in Equation 4, where we introduce A-MoD. We use an average across all attention heads to get an overall importance of tokens which encompasses information across all attention heads instead of individual head which can be used to skip tokens in the transformer block. Moreover, this estimated token importance has a very high correlation (almost 1) with the leave-one-out token importance as shown in Section 4.4.
>
> Q3. We conduct a profiling of the model using the PyTorch profiler and for a further breakdown of the inference time of individual operations as reported in Figure 19 and find that isoFLOP baselines require more GPU time. We attribute this speed up to the fact that A-MoD needs to process fewer overall tokens, in spite of having more depth as MoD layers are placed in every other layer.
>
>
> **Updates to the draft**: As outlined in our general response, we have provided updates to the draft by adding Appendix sections A.8-A.13 along with modifying the abstract as suggested by the reviewer. We also provide Figure 1 to highlight the Pareto-optimality of A-MoD in comparison to MoD and isoFLOP baselines on ImageNet.
>
> 1. **Training from scratch**: Due to the limited time during the discussion phase and the long duration of training from scratch (at least 300 epochs), we are unable to provide isoFLOP runs for training from scratch as of now. However, the experiments are already running and Wwe will update them in the camera ready draft.
>
> 2. **MoD select boundary patches**: The reviewer makes a valid point, that standard routing MoD struggles to choose the correct patches and chooses boundary patches instead which is precisely the motivation of our work. We introduce attention routing which shows improved choosing of tokens as shown in Figure 5.
>
> 3. **IsoFLOP baselines are better on transfer learning tasks in Table 5**: While isoFLOP baselines are slightly better than MoD models (both MoD and A-MoD), Table 5 shows that the MoD architectures are still able to compete with standard isoFLOP baselines. Moreover, as shown in our ablation in Figure 8, A-MoD training is improved by allowing the initial layers to be dense and these A-MoD models are now able to match their isoFLOP counterparts, suggesting that while we show that the standard MoD model with alternate layers skipped are competitive, an improved layer placement of MoDs can enhance their performance further.
>
> 4. **IsoFLOP models**: [1] use isoFLOP models by reducing dimension as well as reducing model depth. We choose to reduce model depth such that we can leverage pretrained weights for isoFLOP models as well. However, we also provide an isoFLOP run for DeiT-S with reduced embedding dimension and training this model from scratch:
>
>     - A-MoD (12.5% Capacity, 2.6GFLOPs): 76.98%
>     - MoD (12.5% Capacity, 2.6GFLOPs): 76.07%
>     - IsoFLOP (2.6GFLOPs): 75.73%
>
> [1] Raposo, David, et al. "Mixture-of-Depths: Dynamically allocating compute in transformer-based language models." arXiv preprint arXiv:2404.02258 (2024).
>
> We are happy to clarify any further questions the reviewer might have and continue this discussion.

---

> > ### Comment · Reviewer_N1Y6 · 2024-12-02
> >
> > Thank you for your efforts and detailed responses. I would like to provide further comments on the following points:
> >
> > **Q2**
> > I still find averaging the attention maps across heads to be an unreasonable approach. To better understand the comparison, I suggest that the authors show the attention maps (or something else) of a vanilla ViT rather than those of MOD or A-MOD.
> >
> > **Q3**
> > The explanation provided does not convince me. With the same FLOPs, ISOFLOP requires fewer layers, whereas A-MOD introduces additional routing operations, branches, etc. This makes it hard to believe that A-MOD can run faster than ISOFLOP. I recommend the authors double-check the experimental settings. If the results are confirmed, it would be helpful to further investigate and provide a more convincing explanation for this result.
> >
> > Additionally, after reviewing the rest of the rebuttal, I noticed that some of my earlier questions remain unaddressed, (e.g., why MoD select boundary patches, which is not reasonable to me, IsoFLOP baselines are better on transfer learning tasks in Table 5).
> >
> > I appreciate the authors' efforts and their engagement in this discussion. However, since most of my concerns remain unresolved, I will maintain my original score. I hope these comments are constructive and can help improve the manuscript.

---

### Official Review · Reviewer_68Jn · 2024-11-03

**Soundness:** 2
**Presentation:** 2
**Contribution:** 2
**Rating:** 5
**Confidence:** 5

**Summary:**

This paper proposes a parameter-free routing method to improve the routing mechanism in the mixture-of-depths (MoD) method. The basic idea is to directly use the attention scores in Transformer.

**Strengths:**

- Using the transformer attention in the MoD routing makes sense.
- Extensive results do prove that the proposed A-MoD outperforms MoD.

**Weaknesses:**

- The entire study is too narrow and limited. This paper is specifically targeted at MoD. However, MoD is just an arxiv paper. Does MoD represent the SoTA in terms of the Pareto frontier? All the experiments are mainly compared with MoD? What about other SoTA methods? BTW, MoD is not impressive in Table 1, where it is even inferior to a simple baseline, isoFLOP.

- Why is higher average attention in (4) corresponding to higher importance? What is the semantic meaning? If so, shouldn't the background tokens in Fig. 4 not be skipped since they have high attentions with many other background tokens? BTW, what are the different columns in Fig. 4 and Fig. 5?

**Questions:**

Please address the issues pointed out in the weakness.

---

> ### Author Response · Authors · 2024-11-22
> **Reply to Reviewer 68Jn**
>
> We thank the reviewer for the valuable feedback.
>
> 1. **Scope of our work and comparison with SOTA**: As stated by the reviewer, the focus of this work is to introduce a new routing method leveraging attention for dynamic computation in Mixture-of-Depth architectures. The results in [1] show that MoD architecture can indeed achieve Pareto-optimal performance. Hence, we focus on improving the training and convergence of MoDs using attention-based routing as successfully shown by our experiments. Yet, we also provide a Pareto-curve for accuracy vs FLOPs in Figure 16 and accuracy vs inference time in Fig 17 that highlights the pareto-optimality of A-MoD.
>     - Regarding Experimental Results:
>       - A-MoD outperforms standard MoD and isoFLOP baselines in 7/8 settings in Table 1.
>       - Please see general response for results on additional experiments, in particular, comparison to state-of-the-art methods on token pruning and merging (See also Figure 16 and Table 6 in the Appendix).
>
> 2. **Regarding attention and token importance, we highlight**:
>     - The score in Equation 4 aggregates the interaction between a token and every other token. Intuitively, this means, that a higher score implies more tokens interact with a given token, and such a token is assigned higher token importance. With this, background tokens do not get higher scores in later layers, as also confirmed in Figure 5.
>     - In Figure 4, each column is an MoD layer for standard routing on top and attention routing on the bottom. We have also updated the figure to highlight the same.
>     - In Figure 5, each column represents an attention head of the last MoD layer in the model for standard routing on top and attention routing on the bottom. We have also updated the figure to highlight the same.
>
> [1] Raposo, David, et al. "Mixture-of-Depths: Dynamically allocating compute in transformer-based language models." arXiv preprint arXiv:2404.02258 (2024).

---

> > ### Comment · Reviewer_68Jn · 2024-11-28
> > **Reviewer response to the author rebuttal**
> >
> > Although I appreciate the authors' rebuttal, unfortunately, it is not convincing. The authors claim MoD is SOTA here while saying they are not claiming global SOTA in the response to Reviewer L4xZ, which is quite confusing. The methodology design is also quite straightforward, lacking innovative components. Some of the rebuttals are unconvincing. For example, regarding Table 1, I was asking "MoD is not impressive ... even inferior to a simple baseline, isoFLOP, while the authors responded with "A-Mod outperforms standard MoD and isoFLOP baselines ...". I will keep my rating.

---

### Official Review · Reviewer_L4xZ · 2024-11-05

**Soundness:** 3
**Presentation:** 3
**Contribution:** 2
**Rating:** 3
**Confidence:** 4

**Summary:**

This paper introduces a parameter-free routing method for mixture-of-depths (MoD) models.
It uses attention maps to discover the importance of each token.
The authors also tested the results on a series of DeiT/ViT models.

**Strengths:**

This paper explores the token importance evaluation through the existing attention maps and thus reduce the overhead of extra layers.

**Weaknesses:**

1. There is no comparison with the current SOTA models since DeiT/ViT are relatively old.
2. The idea is quite similar to the following papers. It would be great to include the comparison (such as accuracy and latency) with these papers.
DynamicViT: Efficient Vision Transformers with Dynamic Token Sparsification and
Token Merging: Your ViT But Faster

**Questions:**

1. what's benefits of the proposed approach compared to traditional ConvNet? It would be great to include some comparison with ResNext etc
2. What's the advantage of the proposed approach against the transformer / convnet hybrid architecture such as LeViT: a Vision Transformer in ConvNet's Clothing for Faster Inference or FastViT: A Fast Hybrid Vision Transformer using Structural Reparameterization

---

> ### Author Response · Authors · 2024-11-22
> **Reply to Reviewer L4xZ**
>
> We thank the reviewer for the valuable feedback.
>
> 1. **No comparison with the current SOTA**: We would like to clarify that the focus of our paper is not to claim a new ‘global’  SOTA method but to improve the field of dynamic computation models, which is still very young but holds great potential.
>    - Here, our main contribution is that A-MoD offers a parameter-free routing mechanism for dynamically skipping tokens, preventing training instabilities and allowing for faster convergence. Since MoDs were designed for transformers and our method is built on leveraging the attention map for routing, we only focus on transformers in this work.
>    - Specifically, we focus on understanding the effects of attention routing on training, convergence and Pareto-front comparisons for standard transformer architectures. Given the success of A-MoD on transformers it can be integrated into other hybrid architectures. Technically, we could also leverage internal representations within CNN to determine a routing. Both points are out of scope for this paper, but we aim to explore this as part of future work.
>     - We also provide a **Pareto-curve of accuracy vs FLOPs in Figure 16** in the Appendix to highlight the pareto-optimality of A-MoD.
>
> 2. **Include the comparison (such as accuracy and latency) with these papers (Dynamic ViT[3], ToMe[2])**: Please see our general response for a comparison with the mentioned existing SOTA methods for the token-pruning and token-merging and for additional experiments to verify the performance of A-MoD. We also provide a Pareto-curve for accuracy vs FLOPs in Figure 16 and accuracy vs inference time in Fig 17 that highlights the pareto-optimality of A-MoD.
>
> [1] Raposo, David, et al. "Mixture-of-Depths: Dynamically allocating compute in transformer-based language models." arXiv preprint arXiv:2404.02258 (2024).
>
> [2] Bolya, Daniel, et al. "Token Merging: Your ViT But Faster." The Eleventh International Conference on Learning Representations.
>
> [3] Rao, Yongming, et al. "Dynamicvit: Efficient vision transformers with dynamic token sparsification." Advances in neural information processing systems 34 (2021): 13937-13949.

---

> > ### Comment · Reviewer_L4xZ · 2024-12-02
> >
> > Thanks a lot for authors' rebuttal and efforts. I have read the response but unfortunately it's not convincing enough for me to change the score.

---

### Author Response · Authors · 2024-11-22
**General response to reviewers**

General Response

We thank all the reviewers for taking the time and providing us with valuable feedback to improve our work.

We would like to highlight the main contributions of this paper as appreciated by the reviewers:

1. Reviewer 68Jn: A-MoD makes sense and extensive results show it is better.
2. Reviewer N1Y6: Experimental results show promising performance.
3. Reviewer je9B: The method is clearly explained and easy to implement.

We would like to provide a summary of our response to address issues raised by the reviewers (L4xZ, N1Y6, je9B) as well as the updates made to the draft. Individual points are addressed in direct responses.  When possible, we have also addressed these points in the paper.

1. **Comparison to more baselines** (Reviewer L4xZ, 68Jn, je9B): We compare our method, A-MoD, to state-of-the-art token merging and token pruning methods including Dynamic ViT [3], A-ViT[4] and Token Merging (ToMe)[2]. A-MoD outperforms Dynamic ViT and A-ViT. Moreover, despite standard training A-MoD is comparable to ToMe which additionally benefits from further training with distillation (see Table 6 in the Appendix). We also provide a Pareto-front curve for FLOPs vs accuracy in Figure 16. Please see Appendix A.8 in our updated draft. We have also stated a summary of the results below:

| **Model**  | **Method**              | **Top-1 Acc (%)** | **FLOPs (G)** |
|------------|-------------------------|-------------------|---------------|
| **DeiT-T** | A-MoD                  | 71.8              | 0.9           |
|            | A-ViT                  | 71.0              | 0.8           |
|            | Dynamic ViT            | 70.9              | 0.9           |
|            | ToMe (with distillation) | 71.69*           | 0.93          |
| **DeiT-S** | A-MoD                  | 78.66             | 3.42          |
|            | A-ViT                  | 78.6              | 3.6           |
|            | Dynamic ViT            | 78.3              | 3.4           |
|            | ToMe (with distillation) | 79.68*           | 3.43          |
*with distillation

2. **Efficiency Results: How many tokens can be dropped without sacrificing performance?** (Reviewer N1Y6): To highlight the efficiency of A-MoD, we compare it with the baseline DeiT-S and report the top-1 accuracy on ImageNet. A-MoD can reduce the number of FLOPs by up to 18% without dropping performance, with standard training and no additional tricks. See Appendix A.11, Table 7. The Pareto-front curve in Figure 16 also confirms the effectiveness of A-MoD.
- DeiT-S baseline (4.6G FLOPS): 79.6%
- A-MoD (3.8G FLOPS): 79.63%

3. **Inference time optimizations** (Reviewer je9B, N1Y6): We acknowledge the reviewer comments that our analysis is limited to FLOPs and not inference time or throughput. We report inference time results for A-MoD below and in Appendix A.10 (Fig. 17) in the updated draft. GPU hardware and low code are not yet optimized for dynamic compute. This is ongoing efforts [5] and we compare the models simply in PyTorch  as is (e.g., no TensorRT, quantization, …).

4. **Training from scratch** (Reviewer N1Y6): We also provide results for training from scratch on ImageNet and observe that A-MoD outperforms standard routing. See Appendix A.12, Table 8.
- DeiT-S after 300 epochs:
  - A-MoD 76.63%
  - MoD 75.9%
- ViT-Base after 160 epochs
  - A-MoD 73.66%
  - MoD 72.47%

5. **A-MoD with Flash Attention** (Reviewer je9B): In response to reviewer je9B, we show that A-MoD can be integrated with Flash Attention without the need to realize the NxN attention matrix explicitly. The detailed derivation is included in Appendix A.8 (Algorithm 1) of our updated draft.

6. **A-MoD beyond classification tasks** (Reviewer 68Jn , je9B): We also provide results with A-MoD on object detection tasks. Preliminary experiments suggest MoD can be applied to object detection with only a marginal drop in performance, with only 50% FLOPs (which has not been done before). The performance of A-MoD is on par with MoD. Longer training runs are in progress and will be updated for the camera ready version. Please see Appendix A.13, Table 9 in the updated draft.

| **Metric**                               | **DETR MOD-50** | **DETR A-MOD-50** | **DETR-Baseline** |
|------------------------------------------|-----------------|-------------------|-------------------|
| **mAP**                                  | 39.6%           | 38.6%             | 39.9%             |
| **GFLOPS (Total)**                       | 83.2            | 83.2              | 86.56             |
| **GFLOPS (Transformer Encoder/Decoder)** | 7.747           | 7.745             | 10.745            |

We hope that our response has addressed the concerns of all the reviewers. We are happy to continue this discussion and further improve our work. Based on our response, we also please request the reviewers to reconsider their scores.

Thank you

Authors

---

> ### Author Response · Authors · 2024-11-22
> **General response continued...**
>
> References
>
> [1] Jain, Gagan, et al. "Mixture of Nested Experts: Adaptive Processing of Visual Tokens." The Thirty-eighth Annual Conference on Neural Information Processing Systems
>
> [2] Bolya, Daniel, et al. "Token Merging: Your ViT But Faster." The Eleventh International Conference on Learning Representations.
>
> [3] Rao, Yongming, et al. "Dynamicvit: Efficient vision transformers with dynamic token sparsification." Advances in neural information processing systems 34 (2021): 13937-13949.
>
> [4] Yin, Hongxu, et al. "A-vit: Adaptive tokens for efficient vision transformer." Proceedings of the IEEE/CVF conference on computer vision and pattern recognition. 2022.
>
> [5] Cao, Shiyi, et al. "MoE-Lightning: High-Throughput MoE Inference on Memory-constrained GPUs." arXiv preprint arXiv:2411.11217 (2024).

---

> ### Author Response · Authors · 2024-11-27
> **Additional updates to draft**
>
> We provide additional updates to the draft, summarized as follows:
>
> 1. We added Figure 1 to the paper to highlight that A-MoD is Pareto-optimal on ImageNet compared to MoD and isoFLOP baselines (Please not that this changes the subsequent Figure numbers accordingly).
> 2. As per the suggestions of Reviewer N1Y6, we conduct a profiling of the model using the PyTorch profiler for inference time of individual operations as reported in Figure 19.

---

### Meta-Review · Area_Chair_dukc · 2024-12-17

**Metareview:**

This paper introduces a parameter-free routing method for mixture-of-depths models. The proposed approach aims to optimize the trade-off between computation and performance, which is especially relevant for large-scale models. The authors demonstrate the effectiveness of their method through experiments on DeiT/ViT models.

However, after the review process, all reviewers voted to reject this submission, with concerns such as limited experiments, insufficient explanation of the experimental results, and a lack of fair comparisons with existing works.

**Additional Comments On Reviewer Discussion:**

During the discussion period between the authors and reviewers, all reviewers responded to the authors' rebuttal and found it unconvincing.

---

### Decision · Program_Chairs · 2025-01-22

Reject